# Efficient Differentiable Contact Model with Long-range Influence

**Xiaohan Ye[1], Kui Wu[2], Taku Komura[1,†], Zherong Pan[3,†]**

[1]The University of Hong Kong [2]LIGHTSPEED [3]Meta

[†]Corresponding authors

## Abstract

With the maturation of differentiable physics, its role in various downstream applications—such as model-predictive control, robotic design optimization, and neural PDE solvers—has become increasingly important. However, the derivative information provided by differentiable simulators can exhibit abrupt changes or vanish altogether, impeding the convergence of gradient-based optimizers. In this work, we demonstrate that such erratic gradient behavior is closely tied to the design of contact models. We further introduce a set of properties that a contact model must satisfy to ensure well-behaved gradient information. Lastly, we present a practical contact model for differentiable rigid-body simulators that satisfies all of these properties while maintaining computational efficiency. Our experiments show that, even from simple initializations, our contact model can discover complex, contact-rich control signals, enabling the successful execution of a range of downstream locomotion and manipulation tasks.

## 1 Introduction

Recent advancements in differentiable physical models (Werling et al., 2021; Huang et al., 2024) have unlocked a range of downstream applications, including model-based reinforcement learning Xu et al. (2022), shooting-based controller optimization (Amos et al., 2018), and robot co-design (Xu et al., 2021). State-of-the-art models now extend to various material types, encompassing both rigid and deformable bodies, while offering first-order gradient information. A notable advantage of differentiable physics is its ability to automatically discover contact-rich motions from trivial initializations (Mordatch et al., 2012; Pang et al., 2023). Achieving this, however, requires an ideal contact model that strikes a balance between accurately approximating physical contact mechanisms and providing meaningful gradient information. Over the years, significant efforts have been made to optimize this balance (Werling et al., 2021; Huang et al., 2024; Le Cleac'h et al., 2023). Despite their significant progress, recent systematic analyses (Antonova et al., 2023; Suh et al., 2022a;b) have highlighted several pitfalls in the gradient information provided by differentiable physics systems. While analytic gradients are beneficial across much of the objective landscape, they can exhibit rugged behavior when optimizing over non-smooth interactions and may vanish in nearly flat regions. Consequently, optimizers are often prone to becoming trapped in undesirable local optima.

To address these challenges, existing techniques (Antonova et al., 2023; Li et al., 2022a) employ global optimization algorithms, such as Bayesian optimization and optimal transportation, to escape local optima. While we agree that global search methods are crucial for complementing local gradient-based optimization, we argue that many of the issues with poor gradient information can be mitigated by improving the contact models within existing differentiable physics frameworks. Specifically, both rugged and vanishing gradients stem from the contact model itself. When two rigid objects come into contact, the abrupt introduction of contact forces results in rugged gradients. Conversely, in the absence of contact, the lack of direct interaction leads to vanishing gradients.

In this paper, we make both theoretical and practical contributions, both aimed at enhancing the gradient landscape of differentiable rigid body simulators. Theoretically, we introduce in Section 3 a set of properties that define a well-behaved contact model. These properties ensure that the con-

tact model supports differentiation and can provably prevent inter-penetration, generate physically plausible contact forces, and provide non-vanishing gradients even when objects are arbitrarily far apart. The last property leads to long-range influence, allowing us to discover novel contact-rich motions, even from a trivial initialization where objects are distant from each other. Practically, we present a computationally tractable contact model in Section 4, which satisfies all these properties as proved in our Appendix A.1 and is applicable to arbitrary articulated bodies represented using triangle meshes. Further in Section 5, we significantly improve the computational efficiency for evaluating the contact model using a Bounding Sphere Hierarchy (BSH). We have incorporated our contact model into a full-featured rigid body simulator and experimented on a row of robotic manipulation and locomotion tasks. Our results show that our method can discover complex, contact-rich control signals from trivial initialization, while previous models can get optimizers stuck at trivial local minima or suffer from slow convergence.

## 2 RELATED WORK

The idea of differentiable physics originates from the pioneering work Todorov (2011), which is then built into the MuJoCo simulator (Tassa et al., 2012), where gradients are computed via costly finite difference schemes. Differentiable simulators are then proposed to use more efficient analytic gradient information. Early works apply this idea to rigid body simulators (de Avila Belbute-Peres et al., 2018) and reduced-order deformable body simulators (Pan & Manocha, 2018). The idea is then adopted in other simulator models (Newbury et al., 2024) for elastic and plastic deformations, articulated bodies, and fluid bodies, to name just a few. Since their invention, differentiable simulators have found many applications in computer graphics, robotics, and machine learning. Early works along this line use differentiable simulators to perform model-predictive control (Tassa et al., 2012) and guide deep policy search (Levine & Koltun, 2013). Differentiable simulators can be combined with appearance models to perform state estimation (Ma et al., 2022) and system identification (Le Lidec et al., 2021). In computer graphics, animators use differentiable simulators to inversely design initial and boundary conditions (Li et al., 2023; Stuyck & Chen, 2023; Du et al., 2021). They can also provide gradient information for physics-informed machine learning such as neural PDE solvers (Heiden et al., 2021) and neural motion planning (Toussaint et al., 2019).

Certain substeps in a simulation procedure are inherently non-differentiable, of which the most important substep is contact handling. It is known that gradient information is lost for collision detection with thin-shell-like objects (Harmon et al., 2009; Li et al., 2022a;b) and the sudden change of contact forces in collision responses incur non-smoothness, which requires manually choosing a specific direction in the Clark subdifferential (Werling et al., 2021). These factors can compromise the gradient information, hindering downstream optimizers' performance. In Suh et al. (2022a), authors propose a mixed gradient-free and gradient-based optimizer to boost the performance of policy search. On a parallel front, Li et al. (2020) showed that the contact model can be reformulated to prevent gradient vanishing for thin-shell-like objects. Differentiable simulator (Huang et al., 2024) built on top of this technique exhibits better robustness. However, we show that even with this technique, optimizers can still suffer from vanishing gradients.

## 3 DIFFERENTIABLE PHYSICS WITH WELL-BEHAVED CONTACT MODEL

In this section, we first formulate the problem of a differentiable physical model, and then formalize the properties of a well-behaved contact model that provides useful gradient information.

### 3.1 DIFFERENTIABLE PHYSICS MODEL

Throughout the paper, we consider articulated bodies geometrically represented using triangle meshes. This is the representation adopted by a majority of differentiable contact models (Werling et al., 2021; Huang et al., 2024; Xu et al., 2021). Formally, we assume the configuration of an articulated body is represented using a set of $V$ vertices located at $x_{1,\cdots,V} \in \mathbb{R}^3$, and we use $x$ without subscript to denote the concatenated vertex vector $x \in \mathbb{R}^{3V}$. The vertices are connected to form a set of $T$ triangles $t_{1,\cdots,T}$, with each cornering three vertices and defined as $t_i = \{i(1), i(2), i(3)\} \subset \{1, \cdots, V\}$. The configuration space $x \in \mathcal{C} \subset \mathbb{R}^{3V}$ can be divided into a

penetrating set $\mathcal{C}_{\text{obs}} = \{x \in \mathcal{C} | \exists t_i \neq t_j \wedge \text{CH}(x_{i(k)\in t_i}) \cap \text{CH}(x_{j(k)\in t_j}) \neq \varnothing\}$, where CH is the closed convex hull of a set of vertices, and a penetration-free set $\mathcal{C}_{\text{free}} = \mathcal{C}/\mathcal{C}_{\text{obs}}$. The goal of a contact model is to impose contact forces on the rigid object to ensure that $x \in \mathcal{C}_{\text{free}}$.

A discrete-time physical model can be cast as a time transition function $x^{t+1} = f(x^t, x^{t-1}, \delta t)$, where we use superscript to denote the time index, $x^t$ is the kinematic state of a body at the $t$th time instance, and $\delta t$ is the timestep size. The concatenation of two kinematic states $\langle x^t, x^{t-1} \rangle$ composes a dynamic state of the body, with velocity approximated as $(x^t - x^{t-1})/\delta t$. It is well-known (Marsden & West, 2001; Gast et al., 2015) that the transition function can be cast as a numerical optimization:

$$x^{t+1} \in \text{argmin}_{x_\star^{t+1}} \mathcal{L}(x_\star^{t+1}, x^t, x^{t-1}, \delta t), \tag{1}$$

which leads to stable performance of modern differentiable position-based dynamics, such as Huang et al. (2024). In these position-based models, the Lagrangian function $\mathcal{L}$ contains various terms that model different behaviors. Specifically, we define:

$$\mathcal{L}(x_\star^{t+1}, x^t, x^{t-1}, \delta t) = \mathcal{I}(x_\star^{t+1}, x^t, x^{t-1}, \delta t) + \mu \mathcal{P}(x_\star^{t+1}) + \mathcal{D}(x_\star^{t+1}, x^t, \delta t),$$

where the term $\mathcal{I}$ models inertial acceleration, $\mathcal{P}$ models the contact potential weighted by a parameter $\mu$, and $\mathcal{D}$ models frictional damping. We refer readers to Huang et al. (2024) for more details of other terms and we focus on the term $\mathcal{P}$ in this paper. In optimization-based simulators, the contact mechanics are mainly determined by the potential $\mathcal{P}$, whose proper definition has been discussed, e.g., in Fisher & Lin (2001); Harmon et al. (2009); Li et al. (2020); Ye et al. (2025).

## 3.2 Properties of Contact Potential

We propose four aspects of indispensable properties of a well-behaved contact potential. **Barrier potential:** It is well-known in the theory of continuous collision handling (Brochu et al., 2012) that a penetrating state $x^t \in \mathcal{C}_{\text{obs}}$ corresponds to non-smooth landscape of the contact potential. Therefore, a well-behaved contact potential should provably prevent any penetrations by ensuring that $x^t \in \mathcal{C}_{\text{free}}$ for any $t$. This property is first proposed and achieved in Harmon et al. (2009), where authors designed $\mathcal{P}$ to be a positive layered potential function that is infinite when $x^t \in \mathcal{C}_{\text{obs}}$. As a result, a numerical optimizer with globalization techniques, such as line search and trust-region, can ensure the monotonic decrease of the Lagrangian $\mathcal{L}$, leading to the finiteness of $\mathcal{P}$ and thus $x^{t+1} \in \mathcal{C}_{\text{free}}$. Li et al. (2020) further elucidates that a potential $\mathcal{P}$ should act as a log-barrier function in interior point optimization, which is formalized as our first property below:

**Property 3.1** (Barrier-Form). *$\mathcal{P}(x) \geq 0$ is continuous for any $x \in \mathcal{C}$ and $\mathcal{P}(x) = \infty$ iff $x \in \mathcal{C}_{obs}$.*

Note that Barrier-Form does not describe the exact contact mechanics, because an exact contact model can only impose contact forces on bodies when they are exactly touching, i.e., $x \in \partial\mathcal{C}_{\text{obs}}$, but our contact potential induces the generalized contact force $-\partial\mathcal{P}/\partial x$ even when $x \notin \mathcal{C}_{\text{obs}}$. To mitigate this issue, Li et al. (2020) proposes to iteratively approximate the true contact mechanics by tuning the coefficient $\mu$. Indeed, we can easily verify that $\lim_{\mu \to 0^+} \mu \mathcal{P}$ converges to the indicator function that equals to $\infty$ if $x \in \partial\mathcal{C}_{\text{obs}}$ and 0 otherwise. **Smoothness:** In order to utilize the primal log-barrier method for computing $x^{t+1}$ by solving optimization would require $\mathcal{P}$ to be at least differentiable. To this end, Li et al. (2020) proposed a differentiable surrogate of the triangle-triangle distance function. Unfortunately, although differentiability is enough for solving Equation 1, it is not enough for providing reliable gradient information. Indeed, the gradient of a numerical optimization takes the following form by the implicit function theorem:

$$\frac{\partial x^{t+1}}{\partial(x^t, x^{t-1})} = -\left[\frac{\partial^2 \mathcal{L}}{\partial x^{t+1^2}}\right]^{-1} \frac{\partial^2 \mathcal{L}}{\partial x^{t+1}\partial(x^t, x^{t-1})},$$

whose proper evaluation requires $\mathcal{P}$ to be twice-differentiable, which is not satisfied in Li et al. (2020); Huang et al. (2024) as shown in our Appendix A.4, leading to our second property:

**Property 3.2** (Smoothness). *$\mathcal{P}$ is twice differentiable at $x \in \mathcal{C}_{free}$.*

Unfortunately, being numerically well-defined does not guarantee that the gradient information can effectively guide the optimizer to find meaningful solutions for downstream applications. To this

end, we introduce two other properties that ensure the contact model is non-prehensile and non-vanishing. **Non-prehensile:** We know that a passive contact can only impose unilateral pushing forces between a pair of contacting objects, instead of pulling objects together. Formally, we introduce a sufficient condition to ensure non-prehensile forces. Let us define two index subsets of well-separated vertices $\mathcal{I} \cap \mathcal{J} = \varnothing$ and $\mathcal{I} \cup \mathcal{J} \subseteq \{1, \cdots, V\}$, such that the convex hull of these sets of vertices are non-overlapping, i.e. $\text{CH}(x_{i \in \mathcal{I}}) \cap \text{CH}(x_{j \in \mathcal{J}}) = \varnothing$. A contact potential between these two subsets can be defined as a pair-wise contact term $\mathcal{P}^{\mathcal{I} \cup \mathcal{J}}(x_{i \in \mathcal{I}}, x_{j \in \mathcal{J}})$ or $\mathcal{P}^{\mathcal{I} \cup \mathcal{J}}$ for short. Note that we can also establish Barrier-Form and Smoothness for a pair-wise contact term $\mathcal{P}^{\mathcal{I} \cup \mathcal{J}}$. To this end, we define $\mathcal{C}_{\text{obs}}^{\mathcal{I} \cup \mathcal{J}} = \{x \in \mathcal{C} | \exists t_i \neq t_j \wedge t_i \cup t_j \subseteq \mathcal{I} \cup \mathcal{J}, \text{CH}(x_{i' \in t_i}) \cap \text{CH}(x_{j' \in t_j}) \neq \varnothing\}$ and $\mathcal{C}_{\text{free}}^{\mathcal{I} \cup \mathcal{J}} = \mathcal{C}/\mathcal{C}_{\text{obs}}^{\mathcal{I} \cup \mathcal{J}}$ and we have the following Barrier-Form and Smoothness for pairwise contact terms:

**Definition 3.1** ( Barrier-Form and Smoothness for pairwise contact terms). *$\mathcal{P}^{\mathcal{I} \cup \mathcal{J}}$ pertains Barrier-Form if $\mathcal{P}^{\mathcal{I} \cup \mathcal{J}} \geq 0$ for any $x \in \mathcal{C}$ and $\mathcal{P}^{\mathcal{I} \cup \mathcal{J}} = \infty$ iff $x \in \mathcal{C}_{\text{obs}}^{\mathcal{I} \cup \mathcal{J}}$. $\mathcal{P}^{\mathcal{I} \cup \mathcal{J}}$ pertains Smoothness if it is twice differentiable at $x \in \mathcal{C}_{\text{free}}^{\mathcal{I} \cup \mathcal{J}}$.*

$\mathcal{P}^{\mathcal{I} \cup \mathcal{J}}$ induces the following contact force on any $x_{i \in \mathcal{I}}$ or $x_{j \in \mathcal{J}}$:

$$f_{i \in \mathcal{I}}^{\mathcal{I} \cup \mathcal{J}} = -\frac{\partial}{\partial x_{i \in \mathcal{I}}} \mathcal{P}^{\mathcal{I} \cup \mathcal{J}}(x_{i \in \mathcal{I}}, x_{j \in \mathcal{J}}) \quad f_{j \in \mathcal{J}}^{\mathcal{I} \cup \mathcal{J}} = -\frac{\partial}{\partial x_{j \in \mathcal{J}}} \mathcal{P}^{\mathcal{I} \cup \mathcal{J}}(x_{i \in \mathcal{I}}, x_{j \in \mathcal{J}}).$$

To allow only non-prehensile forces, we require that each $f_i^{\mathcal{I} \cup \mathcal{J}}$ is pointing from $\text{CH}(x_{j \in \mathcal{J}})$ to $\text{CH}(x_{i \in \mathcal{I}})$ and vice versa. Formally, we define the set of non-zero vectors pointing from $\text{CH}(x_{j \in \mathcal{J}})$ to $\text{CH}(x_{i \in \mathcal{I}})$ as $\mathcal{F}_{\mathcal{J} \to \mathcal{I}} = \{\alpha(a - b) | \alpha > 0 \wedge a \in \text{CH}(x_{i \in \mathcal{I}}) \wedge b \in \text{CH}(x_{j \in \mathcal{J}})\}$, and require that:

$$\forall i \in \mathcal{I} : f_i^{\mathcal{I} \cup \mathcal{J}} \in \mathcal{F}_{\mathcal{J} \to \mathcal{I}} \text{ and } \forall j \in \mathcal{J} : f_j^{\mathcal{I} \cup \mathcal{J}} \in \mathcal{F}_{\mathcal{I} \to \mathcal{J}}. \tag{2}$$

**Non-vanishing:** Our final property is of paramount importance and ensures that a differentiable simulator provides non-zero gradient information at arbitrary configuration. This is ensured by our definition of the non-prehensile force set $\mathcal{F}_{\mathcal{J} \to \mathcal{I}}$. Indeed, since $\text{CH}(x_{j \in \mathcal{J}})$ and $\text{CH}(x_{i \in \mathcal{I}})$ are disjoint, closed convex sets, for any $\alpha(a - b) \in \mathcal{F}_{\mathcal{J} \to \mathcal{I}}$, we have $a \neq b$ and $\alpha > 0$, leading to $f_i^{\mathcal{I} \cup \mathcal{J}} \neq 0$ for all $i \in \mathcal{I}$. We further ensure that the contact forces between every pair of geometric primitives (triangles) are taken into consideration. Put together, we can ensure both properties by requiring that $\mathcal{P}$ is a summation of pairwise contact terms $\mathcal{P}^{\mathcal{I} \cup \mathcal{J}}$ between well-separated vertex clusters:

**Property 3.3** (Non-prehensile & Non-vanishing). *At every $x \in \mathcal{C}_{\text{free}}$, we can define a finite family of set pairs $\mathcal{A}(x) = \{\langle \mathcal{I}, \mathcal{J} \rangle | \mathcal{I} \cap \mathcal{J} = \varnothing \wedge \mathcal{I} \cup \mathcal{J} \subseteq \{1, \cdots, V\} \wedge CH(x_{i \in \mathcal{I}}) \cap CH(x_{j \in \mathcal{J}}) = \varnothing\}$. We have $\mathcal{P} = \sum_{\langle \mathcal{I}, \mathcal{J} \rangle \in \mathcal{A}(x)} \mathcal{P}^{\mathcal{I} \cup \mathcal{J}}$ such that every term $\mathcal{P}^{\mathcal{I} \cup \mathcal{J}}$ satisfy Equation 2, and for every pair of triangles $\langle t_i, t_j \rangle$ on different rigid bodies, we have $t_i \cup t_j \subseteq \mathcal{I} \cup \mathcal{J}$ for at least one $\langle \mathcal{I}, \mathcal{J} \rangle \in \mathcal{A}(x)$.*

This is an important property that allows the gradient information to be provided for arbitrarily distant objects. In many applications, such gradient information can help a local optimizer discover contact-rich motions from trivial initial guesses. Regretfully, we are not aware of any contact model that pertains Barrier-Form, Smoothness, and Non-prehensile & Non-vanishing at the same time. We summarize the failure cases for various properties in Figure 2 and compare the property completeness in Table 1. In Figure 1, we illustrate the main idea behind our contact model in Section 5 that satisfies Non-prehensile & Non-vanishing.

## 4 WELL-BEHAVED CONTACT POTENTIAL

In this section, we propose a practical and well-behaved contact potential. We start by showing that slightly modifying an existing contact potential (Liang et al., 2024; Ye et al., 2025) makes it well-behaved, but such a potential is slow to compute. We then improve its computational efficacy in Section 5 by borrowing ideas from the well-known hierarchical algorithm (Barnes & Hut, 1986) for N-body simulation.

Barrier-Form requires that our contact potential acts as a primal barrier function. However, existing barrier potential functions (Harmon et al., 2009; Li et al., 2020) is derived from a modified triangle-triangle distance function, denoted as $d(x_{i(k) \in t_i}, x_{j(k) \in t_j})$, which is then assembled to form the following contact potential: $\mathcal{P} = \sum_{t_i \neq t_j} P(d(x_{i(k) \in t_i}, x_{j(k) \in t_j}))$, with $P$ being some locally

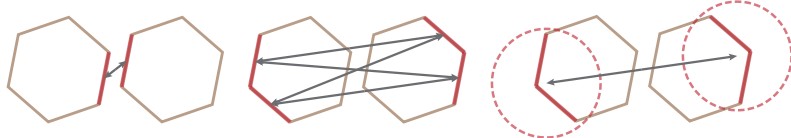

Figure 1: A 2D illustration of our contact force between a pair of hexagons satisfying Non-prehensile & Non-vanishing. Each hexagons have 6 line segments in 2D (resp. triangles in 3D). Left: We compute the exact contact force (arrow) between each pair of nearby line segments. Middle: For faraway pairs of line segments, computing exact contact forces would involve too many segment pairs, e.g. 4 pairs of forces between 2 edges on each hexagon. Right: Instead, we group faraway segments and approximate the contact forces between centers of bounding circles in 2D (resp. bounding spheres in 3D).

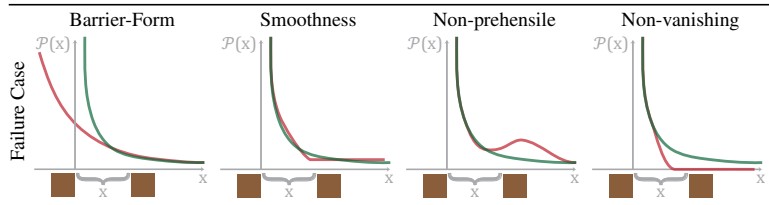

Figure 2: We illustrate failure cases for various properties, where we assume two brown boxes are separated by distance $x$ and plot the contact potential $\mathcal{P}(x)$ that pertains the property in green and fails the property in red. Our Barrier-Form requires $\mathcal{P}$ to tend to infinity as $x \to 0^+$. Smoothness requires $\mathcal{P}$ to have well-defined second-order derivatives. Non-prehensile requires the contact force to always push the two boxes apart. Non-vanishing requires $\mathcal{P}$ and thus the contact force to be non-vanishing for arbitrarily large $x$.

supported barrier function. Unfortunately, this potential is at most first-order differentiable and violates Smoothness and the local support of $P$ violates Non-vanishing. Instead, we propose to adopt the more general contact potential (Liang et al., 2024; Ye et al., 2025) between a pair of convex hulls. These methods use the separating hyperplane theorem to prevent two convex hulls from colliding by inserting a separating hyperplane between them. This hyperplane is then modeled as a physical object with zero-mass, which serves as auxiliary variables to formulate the contact potential. It has been shown that such general contact potential is globally twice differentiable, which serves as a good starting point for our derivation. Since a triangle is convex by nature, the more general contact potential can be adopted to serve as $\mathcal{P}^{t_i \cup t_j}$. Specifically, given a pair of triangles $t_i$ and $t_j$, since the two triangles are both convex sets, we can define a separating plane $p_{ij} = \left( n_{ij}^T, d_{ij} \right)^T \in \mathbb{R}^4$ between them if the two sets are disjoint, with $n_{ij}$ and $d_{ij}$ being the normal and offset, such that: $\langle x_{i(k) \in t_i}, n_{ij} \rangle + d_{ij} > 0$ and $\langle x_{j(k) \in t_j}, n_{ij} \rangle + d_{ij} < 0$. As a result, we introduce the following potential via a nested optimization:

$$\mathcal{P}^{t_i \cup t_j} = \min_{p_{ij}} \mathcal{L}_{ij}(p_{ij}, x_{i(k) \in t_i}, x_{j(k) \in t_j})$$

$$\mathcal{L}_{ij}(p_{ij}, x_{i(k) \in t_i}, x_{j(k) \in t_j}) = \left[ 12 \frac{1}{(1 - \|n_{ij}\|)^+} + \sum_{k=1}^{3} \frac{1}{(\langle x_{i(k)}, n_{ij} \rangle + d_{ij})^+} + \sum_{k=1}^{3} \frac{1}{(\langle x_{j(k)}, -n_{ij} \rangle - d_{ij})^+} \right],$$

where we purposefully introduce a constant coefficient 12 in the first term so that our follow-up derivations take a simpler form, and other positive coefficients can be used as well. As a main point of departure from the original formulation in Liang et al. (2024); Ye et al. (2025), we do not use locally supported log-barrier function, but define the potential function $1/(\bullet)^+ = 1/\max(\bullet, 0)$ that has a global support on $\mathbb{R}^+$ to prevent vanishing gradient. Note that $n_{ij}, d_{ij}$ are computed by minimizing $\mathcal{L}_{ij}$ and thus $n_{ij}$ is not normalized. It is easy to verify that the objective function defined in $\mathcal{P}^{t_i \cup t_j}$ is a strictly convex function with a unique minimizer, so that $\mathcal{P}^{t_i \cup t_j}$ is a well-defined function. With the pair-wise potential defined, we can assemble them and define:

$$\mathcal{P} = \sum_{t_i \neq t_j} \mathcal{P}^{t_i \cup t_j}(x_{i(k) \in t_i}, x_{j(k) \in t_j}), \tag{3}$$

where the summation is taken over triangle pairs on different rigid bodies. We now show that the so-defined contact potential pertains all our desired properties.

| Formulation | Barrier-Form | Smoothness | Non-prehensile | Non-vanishing |
|---|---|---|---|---|
| Turpin et al. (2022); Schwarke et al. (2025) | ✗ | ✗ | ✓ | ✓ |
| Werling et al. (2021); Xu et al. (2022) | ✗ | ✗ | ✓ | ✗ |
| Fisher & Lin (2001); Guendelman et al. (2003) | ✗ | ✗ | ✓ | ✗ |
| Harmon et al. (2009); Li et al. (2020) | ✓ | ✗ | ✓ | ✗ |
| Ye et al. (2025) | ✓ | ✓ | ✓ | ✗ |
| Ours | ✓ | ✓ | ✓ | ✓ |

Table 1: Comparison of property completeness. Contact models based on complementary conditions Werling et al. (2021); Xu et al. (2022) or soft penalty functions Fisher & Lin (2001); Guendelman et al. (2003) cannot guarantee intersection-free or sufficient smoothness. Contact models based on the log-barrier functions Harmon et al. (2009); Li et al. (2020) only have first-order derivatives, which does not support differentiation using the inverse function theorem. Finally, prior contact models have vanishing gradient when the distance is larger than a small margin. Although, Turpin et al. (2022); Schwarke et al. (2025) provides non-vanishing gradients, they still cannot guarantee intersection-free and smoothness, limited by penalty models.

**Lemma 4.1.** *Each pair-wise potential $\mathcal{P}^{t_i \cup t_j}$ in Equation 3 pertains Barrier-Form, Smoothness, and satisfies Equation 2, so that the potential $\mathcal{P}$ pertains Barrier-Form, Smoothness, and Non-prehensile & Non-vanishing.*

At this point, we have shown that Equation 3 is a well-behaved contact potential function. Remarkably, this function is computationally practical. Indeed, each term $\mathcal{P}^{t_i \cup t_j}$ involves a small 4D-optimization problem with a strictly convex objective function, which can be solved efficiently using Newton's method to evaluate $p_{ij}$. The first and second derivatives of $\mathcal{P}^{t_i \cup t_j}$ can then be computed using the inverse function theorem. However, a brute force computation of the potential function $\mathcal{P}$ is not efficient, since it involves terms that account for the contact potential between each pair of disjoint triangles which increases in the square order of triangles.

## 5 Efficient Contact Potential Evaluation

The computational challenge of evaluating Equation 3 lies in accounting for the contact potentials between all pairs of disjoint triangles. This scenario closely parallels the N-body simulation problem, where the forces between all pairs of particles must be computed. Instead of performing a brute-force summation with a computational cost of $O(N^2)$, efficient algorithms such as the tree code (Barnes & Hut, 1986) and the fast multipole expansion (Greengard & Rokhlin, 1987) achieve costs of $O(N \log(N))$ and $O(N)$, respectively. These methods rely on the multipole expansion to separate the influences of source particles from those of target particles. However, several factors make these algorithms unsuitable for our case. First, the multipole expansions for our contact potential $\mathcal{P}^{t_i \cup t_j}$ remain undefined. Second, even if such expansions could be derived, the abrupt transition between the exact potential and its multipole approximation could introduce discontinuities, thereby violating Smoothness. Inspired by the fast multipole method (Greengard & Rokhlin, 1987), we propose instead a modified potential that is also well-behaved and can be evaluated hierarchically. Our main idea is to smoothly transit from the exact potential function $\mathcal{P}^{t_i \cup t_j}$ to simplified functions that can be hierarchically evaluated.

### 5.1 Smooth Transition Between Potentials

Let us consider the pairwise potential between index set $\mathcal{I}$ and $\mathcal{J}$. For an index set $\mathcal{I}$, we define its bounding sphere to be centered at $x_{\mathcal{I}} = \sum_{i \in \mathcal{I}} x_i / |\mathcal{I}|$ with radius $R_{\mathcal{I}} \geq \max_{i \in \mathcal{I}} |x_{\mathcal{I}} - x_i|$. Suppose there are two versions of the potential denoted as $\mathcal{P}_{d_1}^{\mathcal{I} \cup \mathcal{J}}$ and $\mathcal{P}_{d_2}^{\mathcal{I} \cup \mathcal{J}}$, we can smoothly blend the two functions when the distance between $x_{\mathcal{I}}$ and $x_{\mathcal{J}}$ grows from $d_1$ to $d_2$ with $d_1 < d_2$ as illustrated in Figure 3, yielding the following blending potential:

$$\mathcal{P}_{d_1 \to d_2}^{\mathcal{I} \cup \mathcal{J}} = (1 - \phi_{d_1 \to d_2}(x))\mathcal{P}_{d_1}^{\mathcal{I} \cup \mathcal{J}} + \phi_{d_1 \to d_2}(x)\mathcal{P}_{d_2}^{\mathcal{I} \cup \mathcal{J}}, \tag{4}$$

where we define the interpolation function as:

$$\phi_{d_1 \to d_2}(x) = \Phi((\|x_{\mathcal{I}} - x_{\mathcal{J}}\| - d_1)/(d_2 - d_1)) \text{ and } \Phi(d) = \max(\min(6d^5 - 15d^4 + 10d^3, 1), 0).$$

Similar to the tree code algorithm (Barnes & Hut, 1986), our goal of blending is to gradually replace exact potential functions with faster-to-compute approximations. Such blending should not happen when two sets of vertices are too close to each other. In practice, we only allow blending when $R_\mathcal{I} + R_\mathcal{J} \leq d_1$ where $R_\mathcal{I}, R_\mathcal{J}$ are the radii of the bounding spheres of $\mathcal{I}, \mathcal{J}$. We first show the well-behaved nature of potential functions is invariant to blending:

**Lemma 5.1.** *Taking the following assumptions: i)* $R_\mathcal{I} + R_\mathcal{J} \leq d_1 < d_2$; *ii)* $\mathcal{P}_{d_1}^{\mathcal{I} \cup \mathcal{J}}$ *pertains Barrier-Form, Smoothness, and satisfies Equation 2; iii)* $0 \leq \mathcal{P}_{d_2}^{\mathcal{I} \cup \mathcal{J}} \leq \mathcal{P}_{d_1}^{\mathcal{I} \cup \mathcal{J}}$ *when* $\|x_\mathcal{I} - x_\mathcal{J}\| \geq d_1$; *iv)* $\mathcal{P}_{d_2}^{\mathcal{I} \cup \mathcal{J}}$ *has Smoothness, and satisfies Equation 2, then* $\mathcal{P}_{d_1 \to d_2}^{\mathcal{I} \cup \mathcal{J}}$ *has the same properties as* $\mathcal{P}_{d_1}^{\mathcal{I} \cup \mathcal{J}}$.

Lemma 5.1 can be immediately used to blend our potential function $\mathcal{P}^{t_i \cup t_j}$ with a much simpler, closed-form function. Consider moving all three vertices of $t_i$ to the center point $x_{t_i} = (x_{i(1)} + x_{i(2)} + x_{i(3)})/3$ and similarly moving $t_j$ to $x_{t_j}$, then the potential $\mathcal{P}^{t_i \cup t_j}$ takes the following (centered) form after some basic algebraic manipulation:

$$\mathcal{P}_c^{t_i \cup t_j} = \min_{p_{ij}} \left[ 12 \frac{1}{(1 - \|n_{ij}\|)^+} + \sum_{k=1}^{3} \frac{1}{(\langle x_{t_i}, n_{ij} \rangle + d_{ij})^+} + \sum_{k=1}^{3} \frac{1}{(\langle x_{t_j}, -n_{ij} \rangle - d_{ij})^+} \right] = 12 \left[ 1 + \frac{1}{\|x_{t_i} - x_{t_j}\|^{1/2}} \right]^2. \quad (5)$$

We are now ready to apply Lemma 5.1 to blend $\mathcal{P}^{t_i \cup t_j}$ and $\mathcal{P}_c^{t_i \cup t_j}$ in a well-behaved manner:

**Corollary 5.2.** *If we define* $\mathcal{P}_{d_1}^{t_i \cup t_j} = \mathcal{P}^{t_i \cup t_j}$ *and* $\mathcal{P}_{d_2}^{t_i \cup t_j} = \mathcal{P}_c^{t_i \cup t_j}$, *then* $\mathcal{P}_{d_1 \to d_2}^{t_i \cup t_j}$ *pertains Barrier-Form, Smoothness, and satisfies Equation 2.*

Intuitively, Corollary 5.2 allows us to use the exact potential $\mathcal{P}^{t_i \cup t_j}$ when two triangles are very close to each other, while switching to a simpler, closed-form potential $\mathcal{P}_c^{t_i \cup t_j}$ when the centers of two triangles are well-separated by some distance $d_2$.

## 5.2 HIERARCHICAL POTENTIAL BLENDING

In the previous section, we employed blending techniques to smoothly replace the costly contact potential $\mathcal{P}^{t_i \cup t_j}$ with a more computationally efficient closed-form potential $\mathcal{P}_c^{t_i \cup t_j}$. However, since this blending is applied only to individual pairs of triangles, the approach still requires summing over $O(T^2)$ terms. In this section, we fully unlock the potential of blending by hierarchically merging triangles to construct a BSH (Agarwal et al., 2004; Bradshaw & O'Sullivan, 2004) for each rigid body, and then smoothly replace the contact potential of each sphere with a single term. Specifically, we adopt a layered hierarchy, where each sphere

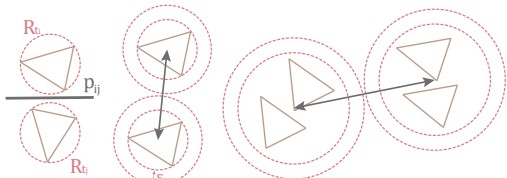

Figure 3: Illustration of our BSH-based contact potential. When two triangles are nearby, we use the exact potential based on separating plane $p_{ij}$ (left). When the center of bounding sphere is separated by at least $(R_{t_i} + R_{t_j})(1 + \epsilon)$, we use the centered potential in Equation 7 (middle). These two cases are combined by smooth blending. The centered potential can be calculated hierarchically for clusters of triangles (right).

tightly encapsulates all the bounding spheres of its two children. Another widely used option is to use a wrapped hierarchy, where each sphere tightly encapsulates the actual geometry. Although wrapped hierarchy achieves tighter bound, the layered hierarchy is required in our method to ensure well-behaved properties. Our BSH is defined below:

**Definition 5.3.** *A BSH is a binary tree, where each node contains an index subset of vertices* $\mathcal{I} \subseteq \{1, \cdots, N\}$ *that is the union of the two subsets of its left and right children, denoted as* $\mathcal{I} = \mathcal{I}_l \cup \mathcal{I}_r$. *The radius* $R_\mathcal{I}$ *is the smallest radius encapsulating the bounding spheres of two children. Each leaf node stores a single triangle* $t_i$. *Further, each node's sphere is centered at* $x_\mathcal{I}$ *with radius* $R_\mathcal{I}$.

Throughout the paper, we use the associated index subset $\mathcal{I}$ to refer to a BSH node. There are many ways to practically construct our BSH, mostly using a greedy algorithm to iteratively merge nodes, and we adopt the technique of Bradshaw & O'Sullivan (2004). Given the BSH, we propose a recursive definition of our contact potential $\mathcal{P}_{\text{BSH}}^{\mathcal{I} \cup \mathcal{J}}$, one for each node pair $\mathcal{I}$ and $\mathcal{J}$ on two rigid bodies, and use the potential of the root nodes as our final contact potential. Finally, we show that our definition pertains Barrier-Form, Smoothness, and Non-prehensile & Non-vanishing. We start

from the base case. For a pair of leaf nodes $t_i \cup t_j$, we use Corollary 5.2 to define the following pairwise potential:

$$\begin{cases} d_1 = R_{t_i} + R_{t_j} \text{ and } d_2 = (1 + \epsilon)d_1 \\ \mathcal{P}_{d_1}^{t_i \cup t_j} = \mathcal{P}^{t_i \cup t_j} \text{ and } \mathcal{P}_{d_2}^{t_i \cup t_j} = \mathcal{P}_c^{t_i \cup t_j} \\ \mathcal{P}_{\text{BSH}}^{t_i \cup t_j} = \mathcal{P}_{d_1 \to d_2}^{t_i \cup t_j} \end{cases} . \tag{6}$$

Specifically, we blend the exact potential between the pair of triangles and the closed-form centered potential, when the distance between triangle centers grows by a factor of $\epsilon$. We leave $\epsilon$ as a user-defined margin that controls the exactness of potential evaluation. Next, given an arbitrary internal node with two child nodes being $\mathcal{I}$ and $\mathcal{J}$, we recursively replace the more accurate potential between child nodes with a single potential between parent nodes. Let us suppose $\mathcal{I} \cap \mathcal{J} = \varnothing$, we define a potential of similar form as Equation 5:

$$\mathcal{P}_c^{\mathcal{I} \cap \mathcal{J}} = 12 \left[ 1 + 1/\sqrt{\|x_{\mathcal{I}} - x_{\mathcal{J}}\|} \right]^2, \tag{7}$$

which is the potential penalizing distance between two sphere centers. We can only use the centered potential when the two spheres are well-separated, i.e. $R_{\mathcal{I}} + R_{\mathcal{J}} \leq \|x_{\mathcal{I}} - x_{\mathcal{J}}\|$. Otherwise, we have to use the more accurate potential by descending the tree and sum up the pair-wise terms between each pair of child nodes. Specifically, we define the set of child nodes as $C(\mathcal{I}) = \{\mathcal{I}_l, \mathcal{I}_r\}$ if $\mathcal{I}$ is an internal node and $C(\mathcal{I}) = \{\mathcal{I}\}$ if $\mathcal{I}$ is a leaf node. Finally, we define the following potential between internal node:

$$\begin{cases} d_1 = R_{\mathcal{I}} + R_{\mathcal{J}} \text{ and } d_2 = (1 + \epsilon)d_1 \\ \mathcal{P}_{d_1}^{\mathcal{I} \cup \mathcal{J}} = \sum_{\mathcal{I}_c \in C(\mathcal{I})} \sum_{\mathcal{J}_c \in C(\mathcal{J})} \mathcal{P}_{\text{BSH}}^{\mathcal{I}_c \cup \mathcal{J}_c} \text{ and } \mathcal{P}_{d_2}^{\mathcal{I} \cup \mathcal{J}} = \mathcal{P}_c^{\mathcal{I} \cup \mathcal{J}} \\ \mathcal{P}_{\text{BSH}}^{\mathcal{I} \cup \mathcal{J}} = \mathcal{P}_{d_1 \to d_2}^{\mathcal{I} \cup \mathcal{J}} \end{cases} . \tag{8}$$

The main idea behind our formulation is illustrated in Figure 3 and we are ready to present our main result, which shows that the so-defined contact potential is well-behaved:

**Theorem 5.4.** *If $\epsilon > 0$ then $\mathcal{P} = \sum_{\mathcal{I} \neq \mathcal{J}} \mathcal{P}_{BSH}^{\mathcal{I} \cup \mathcal{J}}$ pertains Barrier-Form, Smoothness, and Non-prehensile & Non-vanishing, where the summation is taken over the root nodes of different rigid bodies.*

This result lays the foundation for our efficient-to-evaluate and well-behaved contact model. Although analyzing the cost of evaluating $\mathcal{P}$ could be rather difficult for general cases, we follow the idea of fast multiple expansion (Greengard & Rokhlin, 1987) and analyze the cost of evaluating the contact potential for a uniform grid, where we show in Appendix A.2 that the cost is $O(T)$. Finally, we notice that a contact model should account for frictional contact forces. In Appendix A.3, we show that the frictional damping potential proposed in Ye et al. (2025) can be slightly extended to ours.

## 6 EVALUATION

We evaluated our method in a row of five contact-rich manipulation and locomotion tasks: Billiards, Push, Sort, Ant-Push and Gather. We optimize the sequence of control signals using gradient descent at a fixed learning rate to minimize user-defined loss functions. More experiments and details are deferred to Appendix A.6. For fairness, we compare our contact model with the standard IPC model used in Li et al. (2020); Huang et al. (2024), and SDRS contact model proposed by Ye et al. (2025), which only violates Non-vanishing. We also compare with MuJoCo simulator (Tassa et al., 2012), which uses soft contact and provides gradient by finite-difference schemes. Finally, we compare with Suh et al. (2022b) that uses first-order bundled gradient.

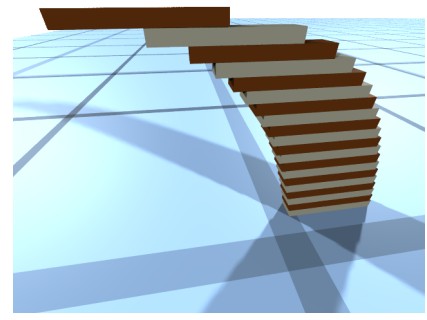

Figure 4: Book stacking problem.

**Physics Accuracy:** Due to our method generating contact forces between objects that are not in direct contact, we need to validate the physical accuracy of our contact model under different contact coefficients $\mu$. We validated the contact model's physical accuracy through the book stacking problem (Hall, 2005) in Figure 4. In this scenario, we stacked 20 planks with dimensions of $2.0m \times 0.4m \times 0.2m$ and mass of 0.16kg sequentially, extending each plank outward to the maximum theoretical distance from the bottom plank without collapsing. To account for simulation errors, they were shifted inward by 0.1% of the plank's length. We verified whether the system could remain stable under different $\mu$. Subsequently, we measured the margin between each plank under different $\mu$ and calculated the error between the contact force the top plank received from adjacent planks and the theoretical value. The results, shown in Table 2, indicate that the system remains stable when $\mu < 1e^{-6}$, with margin errors in the millimeter range, and the errors between contact forces and theoretical values are negligible.

| Contact Coefficient $\mu$ | $1e^{-5}$ | $1e^{-6}$ | $1e^{-7}$ | $1e^{-8}$ | $1e^{-10}$ |
|---|---|---|---|---|---|
| Margin (m) | $1.82e^{-2}$ | $5.67e^{-3}$ | $1.47e^{-3}$ | $4.12e^{-4}$ | $3.13e^{-5}$ |
| Contact force (N) | $6.38e^{-3}$ | $6.40e^{-4}$ | $6.42e^{-5}$ | $6.42e^{-6}$ | $6.42e^{-8}$ |
| Success | ✗ | ✓ | ✓ | ✓ | ✓ |

Table 2: The relationship between the margin between the planks, the error in the contact force received by the top plank, and the stability of the system with respect to the contact coefficient $\mu$ in the book stacking problem.

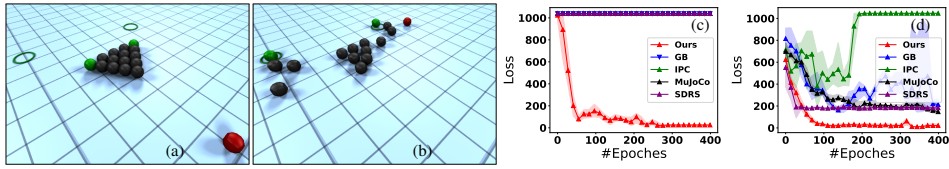

Figure 5: We show the initial (a) and final (b) frame of the billiards task, the convergence history uses trivial initialization (c) and random sampling initialization (d).

**Billiards:** In this benchmark, we have 16 balls on the ground with one target red ball whose initial horizontal position and velocity can be controlled. The goal of control is for the two green balls to reach the target positions (green circle), where the loss function is the squared distance between the green balls and the center of green circles. We experiment with two different methods for setting initial solutions. Our first method uses trivial initialization where certain rigid objects in a scenario are far apart, for which the gradient might vanish except our method. Our second method uses random sampling of control signals to find an initial solution for which gradient information does not vanish. The convergence history of various contact models and initialization strategies is summarized in Figure 5. We optimize a trajectory with 100 timesteps at a timestep size of 0.04. Except our method, other methods cannot make any progress without sampling due to gradient vanishing, which can be fixed via sampling, while our method achieves faster convergence, with or without sampling. We further pick random frames in this scenario and compare the cost of computing the contact potential using our method and IPC in Figure 6. Clearly, our method is slower than IPC due to nested optimization and hierarchical blending. But our method accelerated by BSH is much faster than the brute-force version computing all pairwise potentials between triangles.

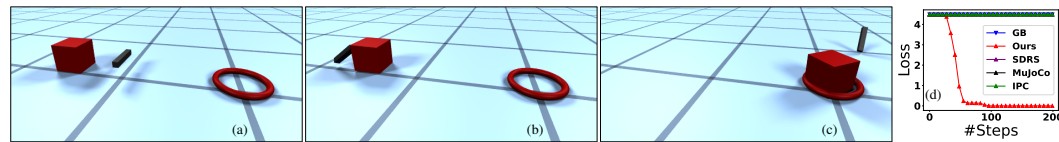

Figure 7: We show the initial (a), middle (b), final (c) frame of the push task, and the convergence history (d) of our method, IPC-based contact model, SDRS, MuJoCo and gradient bundle method (GB).

**Push:** In this benchmark, we optimize the position and orientation of a rod to push the red box to reach a target red circle, where the loss is the squared distance between the box and the red circle. For this benchmark, we use receding-horizon control to generate a trajectory of 200 frames with a horizon of only 48 frames, i.e., we iteratively optimize a 48-frame

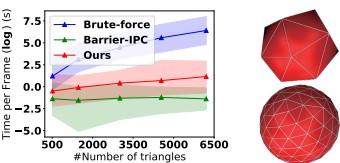

Figure 6: The cost of computing contact potential (left) under different ball mesh resolutions (right).

sub-trajectory and only apply the first action. The rod is initially located in front of the box, and our contact model guides the rod to first move around to the back of the box, and then push it multiple times to reach the target area as illustrated in Figure 7. This result demonstrates the ability of our contact model to discover multi-stage, contact-rich motions from a trivial initial guess. In comparison, all other methods make no progress, even with random sampling to find an initial solution that gradient does not vanish. **Gather & Sort:** In these two benchmarks, we further increase the

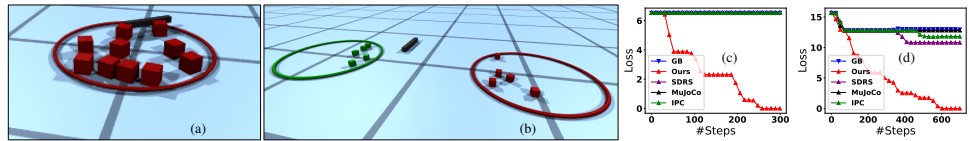

Figure 8: The final frame of the gather (a) and sort (b) task, with convergence history in (c) and (d).

complexity to have the rod manipulate multiple objects on the table. In the gather task, 10 cubes are disseminated and to be pushed together into the target region. In the sort task, 10 cubes are randomly labeled and to be pushed into separate regions according to their labels. The loss is the sum of squared distance between boxes and targets. The frames and convergence history are illustrated in Figure 8. Again, other methods make little to no progress, while our method successfully accomplishes the task, relying solely on the gradient information. **Ant-Push:** In this benchmark, we

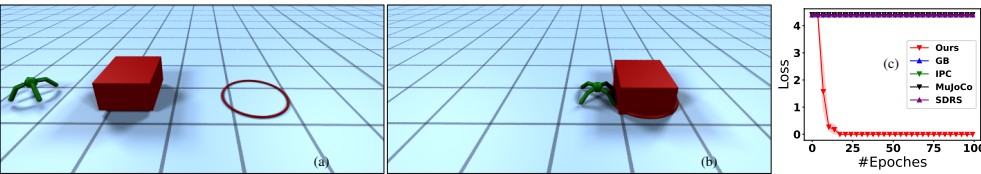

Figure 9: The initial (a), final (b) frame of the Ant-Push task, and the convergence history (c).

show that our method can work with articulated robot models. We optimize for a 9-link, 16-DOF ant robot to push a box to the target position. Once again, our method significantly outperforms other methods, as illustrated int Figure 9.

# 7    CONCLUSION

We present a detailed analysis of the qualifications for a contact model to be well-behaved, which strictly prevents collisions, supports differentiable simulations, induce non-prehensile forces, and avoids vanishing gradients. By hierarchically evaluating the contact potentials assisted by a BSH, we further present a well-behaved contact model that is also efficient to evaluate. By analysis on the special case of a uniform grid, we show that the complexity of evaluating our contact potential is linear. Through evaluations on various motion planning and control tasks, we highlight that our model can guide a gradient-based optimizer to search for complex motion plans and locomotion gaits that are impossible for previous contact models. Our method is not without its problems. First, we can only handle rigid bodies, and we cannot deal with more general deformable objects for soft robot locomotion or soft object manipulation. This is because our bounding spheres might not bound the actual triangles if deformation happens, which could potentially violate Non-prehensile & Non-vanishing. Second, our contact potential involves a recursive definition and requires a nested optimization between pairs of triangles, which incurs considerable overhead to a conventional rigid body simulator.

ACKNOWLEDGMENT

This project was partially funded by Meta. It was also partially funded by the Research Grants Council of Hong Kong (Ref: 17210222), and by the Innovation and Technology Commission of the HKSAR Government under the ITSP-Platform grant (Ref: ITS/335/23FP) and the InnoHK initiative (TransGP project). Part of the research was conducted in the JC STEM Lab of Robotics for Soft Materials, funded by The Hong Kong Jockey Club Charities Trust.

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

# A  APPENDIX

## A.1  ADDITIONAL PROOFS

*proof of Lemma 4.1.*  **Barrier-Form** If $x \in \mathcal{C}_{\text{free}}$ then each pair of $t_i, t_j$ on different rigid bodies is disjoint as the convex hulls are closed sets. Therefore, by the separating plane theorem, there exists a separating plane $p_{ij}$ and some positive $\epsilon_{ij} > 0$ such that:

$$\|n_{ij}\| = 1 \wedge \langle x_{i(k)\in t_i}, n_{ij} \rangle + d_{ij} \geq \epsilon_{ij}/2 \wedge \langle x_{j(k)\in t_j}, n_{ij} \rangle + d_{ij} \leq -\epsilon_{ij}/2.$$

Clearly, $\mathcal{P}^{t_i \cup t_j} \leq \mathcal{L}_{ij}(p_{ij}/2, x_{i(k)\in t_i}, x_{j(k)\in t_j}) < \infty$. On the other hand, if $x \in \mathcal{C}_{\text{obs}}$, then there exists a non-disjoint pair $t_i, t_j$, so for any separating plane $p_{ij}$ there exists $\langle x_{i(k)}, n_{ij} \rangle + d_{ij} \leq 0$ or $x_{j(k)}, n_{ij} \rangle + d_{ij} \geq 0$, leading to $\mathcal{P}^{t_i \cup t_j} = \infty$. Finally, at any feasible solution $p_{ij}$, we must have $\|n_{ij}\| > 0$ because otherwise, we have:

$$\mathcal{L}_{ij} = \sum_{k=1}^{3} \frac{1}{(d_{ij})^+} + \sum_{k=1}^{3} \frac{1}{(-d_{ij})^+} = \infty.$$

We have thus established  Barrier-Form for each $\mathcal{P}^{t_i \cup t_j}$ and thus $\mathcal{P}$.

**Smoothness** This follows from the inverse function theorem (Dontchev & Rockafellar, 2009), the smoothness of problem data $\mathcal{L}_{ij}$, and strictly convexity of $\mathcal{L}_{ij}$.

**Non-prehensile & Non-vanishing** We can simply define $\mathcal{A}(x) = \{\langle t_i, t_j \rangle | t_i \neq t_j\}$ such that every pair of disjoint triangles $\langle t_i, t_j \rangle$ appears in exactly one term of $\mathcal{P}^{t_i \cup t_j}$. Thus, we only need to verify that $f_{i(k)\in t_i}^{t_i \cup t_j} \in \mathcal{F}_{t_j \to t_i}$ and the case with $f_{j(k)\in t_j}^{t_i \cup t_j}$ is symmetric. By the implicit function theorem, we can derive the analytic formula:

$$f_{i(k)\in t_i}^{t_i \cup t_j} = \frac{n_{ij}}{(\langle x_{i(k)}, n_{ij} \rangle + d_{ij})^2}.$$

Suppose $n_{ij} = \alpha(a - b)$ for some $a \in \text{CH}(x_{i(k)\in t_i})$, $b \in \text{CH}(x_{j(k)\in t_j})$, and $\alpha > 0$, then we have $f_{i(k)\in t_i}^{t_i \cup t_j} = \alpha(a - b)/(\langle x_{i(k)}, n_{ij} \rangle + d_{ij})^2 \in \mathcal{F}_{t_j \to t_i}$. Therefore, we can in turn prove the sufficient condition that $n_{ij} = \alpha(a - b)$. Due to the optimality of $\mathcal{L}_{ij}$ with respect to $p_{ij}$, we have:

$$0 = \frac{\partial \mathcal{L}_{ij}}{\partial n_{ij}} = 12 \frac{n_{ij}}{\|n_{ij}\|(1 - \|n_{ij}\|)^2} - \sum_{k=1}^{3} \frac{x_{i(k)}}{(\langle x_{i(k)}, n_{ij} \rangle + d_{ij})^2} + \sum_{k=1}^{3} \frac{x_{j(k)}}{(\langle x_{j(k)}, -n_{ij} \rangle - d_{ij})^2}$$

$$0 = \frac{\partial \mathcal{L}_{ij}}{\partial d_{ij}} = - \sum_{k=1}^{3} \frac{1}{(\langle x_{i(k)}, n_{ij} \rangle + d_{ij})^2} + \sum_{k=1}^{3} \frac{1}{(\langle x_{j(k)}, -n_{ij} \rangle - d_{ij})^2}.$$

From the above two equations, we can conclude that $n_{ij} = \alpha(a - b)$ by defining:

$$\alpha = \frac{\|n_{ij}\|(1 - \|n_{ij}\|)}{12} \sum_{k=1}^{3} \frac{1}{(\langle x_{i(k)}, n_{ij} \rangle + d_{ij})^2} > 0$$

$$a = \left[ \sum_{k=1}^{3} \frac{x_{i(k)}}{(\langle x_{i(k)}, n_{ij} \rangle + d_{ij})^2} \right] / \left[ \sum_{k=1}^{3} \frac{1}{(\langle x_{i(k)}, n_{ij} \rangle + d_{ij})^2} \right] \in \text{CH}(x_{i(k)\in t_i})$$

$$b = \left[ \sum_{k=1}^{3} \frac{x_{j(k)}}{(\langle x_{j(k)}, -n_{ij} \rangle - d_{ij})^2} \right] / \left[ \sum_{k=1}^{3} \frac{1}{(\langle x_{j(k)}, -n_{ij} \rangle - d_{ij})^2} \right] \in \text{CH}(x_{j(k)\in t_j}),$$

thus all is proved. $\square$

*Proof of Lemma 5.1.*  **Barrier-Form** Case I: When $\|x_{\mathcal{I}} - x_{\mathcal{J}}\| < d_1$, $\mathcal{P}_{d_1 \to d_2}^{\mathcal{I} \cup \mathcal{J}} = \mathcal{P}_{d_1}^{\mathcal{I} \cup \mathcal{J}}$ which pertains  Barrier-Form since $\mathcal{P}_{d_1}^{\mathcal{I} \cup \mathcal{J}}$ pertains  Barrier-Form by assumption. Case II: When $\|x_{\mathcal{I}} - x_{\mathcal{J}}\| = d_1 \geq R_{\mathcal{I}} + R_{\mathcal{J}}$, there are two sub-cases. Case II.a: If $x \in \mathcal{C}_{\text{obs}}^{\mathcal{I} \cup \mathcal{J}}$, e.g. where two bounding spheres are just touching and the touching point lies on a common triangle, then $0 \leq \mathcal{P}_{d_2}^{\mathcal{I} \cup \mathcal{J}} \leq \mathcal{P}_{d_1}^{\mathcal{I} \cup \mathcal{J}} = \infty$ where the first two inequalities are due to our assumption and the last equality is due to $\mathcal{P}_{d_1}^{\mathcal{I} \cup \mathcal{J}}$ pertaining  Barrier-Form, so $\mathcal{P}_{d_1 \to d_2}^{\mathcal{I} \cup \mathcal{J}} = \mathcal{P}_{d_1}^{\mathcal{I} \cup \mathcal{J}} = \infty$. Case II.b: If $x \in \mathcal{C}_{\text{free}}^{\mathcal{I} \cup \mathcal{J}}$, then we have

$0 \le \mathcal{P}_{d_2}^{\mathcal{I} \cup \mathcal{J}} \le \mathcal{P}_{d_1}^{\mathcal{I} \cup \mathcal{J}} < \infty$ following the same reasoning as case II.a, so $0 \le \mathcal{P}_{d_1 \to d_2}^{\mathcal{I} \cup \mathcal{J}} < \infty$. Case III: When $\|x_{\mathcal{I}} - x_{\mathcal{J}}\| > d_1 \ge R_{\mathcal{I}} + R_{\mathcal{J}}$, then we must have $x \in \mathcal{C}_{\text{free}}^{\mathcal{I} \cup \mathcal{J}}$ and the same analysis as case II.b leads to $0 \le \mathcal{P}_{d_1 \to d_2}^{\mathcal{I} \cup \mathcal{J}} < \infty$. Thus, we have verified that $\mathcal{P}_{d_1 \to d_2}^{\mathcal{I} \cup \mathcal{J}}$ pertains Barrier-Form in all cases.

**Smoothness** This is due to Smoothness in $\mathcal{P}_{d_1}^{\mathcal{I} \cup \mathcal{J}}, \mathcal{P}_{d_2}^{\mathcal{I} \cup \mathcal{J}}$, and the second differentiability of $\phi_{d_1 \to d_2}$.

**Non-prehensile & Non-vanishing** The force $f_{i \in \mathcal{I}}^{\mathcal{I} \cup \mathcal{J}}$ induced by $\mathcal{P}_{d_1 \to d_2}^{\mathcal{I} \cup \mathcal{J}}$ takes the following form:

$$
f_{i \in \mathcal{I}}^{\mathcal{I} \cup \mathcal{J}} = \underbrace{-(1 - \phi_{d_1 \to d_2}(x)) \frac{\partial \mathcal{P}_{d_1}^{\mathcal{I} \cup \mathcal{J}}}{\partial x_{i \in \mathcal{I}}}}_{\text{term I}} \underbrace{-\phi_{d_1 \to d_2}(x) \frac{\partial \mathcal{P}_{d_2}^{\mathcal{I} \cup \mathcal{J}}}{\partial x_{i \in \mathcal{I}}}}_{\text{term II}}
$$
$$
\underbrace{-(\mathcal{P}_{d_2}^{\mathcal{I} \cup \mathcal{J}} - \mathcal{P}_{d_1}^{\mathcal{I} \cup \mathcal{J}}) \frac{\phi'((\|x_{\mathcal{I}} - x_{\mathcal{J}}\| - d_1)/(d_2 - d_1))}{(d_2 - d_1)\|x_{\mathcal{I}} - x_{\mathcal{J}}\| |\mathcal{I}|}(x_{\mathcal{I}} - x_{\mathcal{J}})}_{\text{term III}}.
$$
$$(9)$$

There are three terms and we show that each term belongs to $\mathcal{F}_{\mathcal{J} \to \mathcal{I}}$. For term I, we know that $-\partial \mathcal{P}_{d_1}^{\mathcal{I} \cup \mathcal{J}}/\partial x_{i \in \mathcal{I}} \in \mathcal{F}_{\mathcal{J} \to \mathcal{I}}$ since $\mathcal{P}_{d_1}^{\mathcal{I} \cup \mathcal{J}}$ satisfies Non-prehensile & Non-vanishing. Since the coefficient $(1 - \phi_{d_1 \to d_2}(x)) \ge 0$ and $\mathcal{F}_{\mathcal{J} \to \mathcal{I}}$ is a cone, we conclude that term I is zero or belongs to $\mathcal{F}_{\mathcal{J} \to \mathcal{I}}$. The same reasoning applies to term II. For term III, $x_{\mathcal{I}} - x_{\mathcal{J}}$ belongs to $\mathcal{F}_{\mathcal{J} \to \mathcal{I}}$. Since $\mathcal{P}_{d_2}^{\mathcal{I} \cup \mathcal{J}} \le \mathcal{P}_{d_1}^{\mathcal{I} \cup \mathcal{J}}$ by our assumption, the remaining coefficient is non-negative, thus term III is zero or belongs to $\mathcal{F}_{\mathcal{J} \to \mathcal{I}}$. Finally, at least one of term I or term II is non-zero, so we conclude that $f_{i \in \mathcal{I}}^{\mathcal{I} \cup \mathcal{J}} \in \mathcal{F}_{\mathcal{J} \to \mathcal{I}}$ and all is proved. $\square$

*Proof of Corollary 5.2.* We first show that Equation 5 is correct. The objective function in Equation 5 is derived by replacing all $x_{i(k)}$ and $x_{j(k)}$ in $\mathcal{L}_{ij}$ with the center points $x_{t_i}$ and $x_{t_j}$, respectively. By symmetry, the optimal separating plane must be the middle surface between $x_{t_i}$ and $x_{t_j}$, taking the following form:

$$ n_{ij} = \alpha(x_{t_i} - x_{t_j}) \text{ and } d_{ij} = -\alpha/2 \langle x_{t_i} - x_{t_j}, x_{t_i} + x_{t_j} \rangle. $$

Plugging the optimal separating plane and solving for $\alpha$ leads to Equation 5. Next, we show that all three assumptions in Lemma 5.1 hold. In fact, there are only two non-trivial assumptions. We first show that $\mathcal{P}_c^{t_i \cup t_j} \le \mathcal{P}^{t_i \cup t_j}$ when $\|x_{\mathcal{I}} - x_{\mathcal{J}}\| \ge d_1$. Let use denote $p_{ij}^\star$ as the optimal separating plane for $\mathcal{P}^{t_i \cup t_j}$, then we have the following inequality:

$$
\mathcal{P}_c^{t_i \cup t_j} = \operatorname*{argmin}_{p_{ij}} \left[ 12 \frac{1}{(1 - \|n_{ij}\|)^+} + \sum_{k=1}^3 \frac{1}{(\langle x_{t_i}, n_{ij} \rangle + d_{ij})^+} + \sum_{k=1}^3 \frac{1}{(\langle x_{t_j}, -n_{ij} \rangle - d_{ij})^+} \right]
$$
$$
\le 12 \frac{1}{(1 - \|n_{ij}^\star\|)^+} + \sum_{k=1}^3 \frac{1}{(\langle x_{t_i}, n_{ij}^\star \rangle + d_{ij}^\star)^+} + \sum_{k=1}^3 \frac{1}{(\langle x_{t_j}, -n_{ij}^\star \rangle - d_{ij}^\star)^+}
$$
$$
\le 12 \frac{1}{(1 - \|n_{ij}^\star\|)^+} + \sum_{k=1}^3 \frac{1}{(\langle x_{i(k)}, n_{ij}^\star \rangle + d_{ij}^\star)^+} + \sum_{k=1}^3 \frac{1}{(\langle x_{j(k)}, -n_{ij}^\star \rangle - d_{ij}^\star)^+}
$$
$$
= \mathcal{P}^{t_i \cup t_j},
$$

where the first inequality is due to optimality of $\mathcal{P}_c^{t_i \cup t_j}$, the second inequality is due to the convexity of function $1/(\langle \bullet, n_{ij}^\star \rangle + d_{ij}^\star)^+$ and $1/(\langle \bullet, -n_{ij}^\star \rangle - d_{ij}^\star)^+$. We then show that $\mathcal{P}_c^{t_i \cup t_j}$ satisfies Equation 2. The force on any $x_{i(k)}$ takes the following form:

$$
f_{i \in \mathcal{I}}^{\mathcal{I} \cup \mathcal{J}} = \frac{4(x_{t_i} - x_{t_j})}{\|x_{t_i} - x_{t_j}\|^{5/2}} \left[ 1 + \frac{1}{\|x_{t_i} - x_{t_j}\|^{1/2}} \right],
$$

which clearly belongs to $\mathcal{F}_{\mathcal{J} \to \mathcal{I}}$, thus all is proved. $\square$

**Lemma A.1.** *If $\epsilon > 0$ then $\mathcal{P}_{\text{BSH}}^{\mathcal{I} \cup \mathcal{J}}$ pertains Barrier-Form for any node pair $\mathcal{I}, \mathcal{J}$ of two rigid bodies.*

*Proof.* First, by induction from leaf to the root node, we can verify that $0 \le \mathcal{P}_{\text{BSH}}^{\mathcal{I} \cup \mathcal{J}}$. Second, suppose $x \in \mathcal{C}_{\text{free}}$, then all the pair-wise terms between leaf nodes $\mathcal{P}_{\text{BSH}}^{t_i \cup t_j} < \infty$. Further, for all the centered

potential in Equation 7, we have $\mathcal{P}_c^{\mathcal{I} \cup \mathcal{J}} < \infty$ because they are evaluated only when $R_{\mathcal{I}} + R_{\mathcal{J}} \leq \|x_{\mathcal{I}} - x_{\mathcal{J}}\|$. The root potential $\mathcal{P}_{\text{BSH}}^{\mathcal{I} \cup \mathcal{J}}$ is then derived by a finite number of blending and summation so we have $\mathcal{P}_{\text{BSH}}^{\mathcal{I} \cup \mathcal{J}} < \infty$. Third, at any $x \in \mathcal{C}_{\text{obs}}$, there exists a non-disjoint pair $t_i, t_j$ belonging to the two rigid bodies. We will show the following two claims hold by induction from leaf to root:

- $\mathcal{P}_{\text{BSH}}^{\mathcal{I} \cup \mathcal{J}} = \infty$ at any node such that $t_i \subseteq \mathcal{I}$ and $t_j \subseteq \mathcal{J}$.

**Base Step:** $\mathcal{P}_{\text{BSH}}^{t_i \cup t_j} = \mathcal{P}_{d_1 \to d_2}^{t_i \cup t_j} = \infty$ by Corollary 5.2.

**Inductive Step:** We assume our first claim holds for any $t_i \in \mathcal{I}_c \in C(\mathcal{I})$ and $t_j \in \mathcal{J}_c \in C(\mathcal{J})$. If $t_i \subseteq \mathcal{I}$ and $t_j \subseteq \mathcal{J}$, we must have $\|x_{\mathcal{I}} - x_{\mathcal{J}}\| \leq R_{\mathcal{I}} + R_{\mathcal{J}}$ due to the pair $t_i, t_j$ being non-disjoint. The children set satisfying $t_i \in \mathcal{I}_c \in C(\mathcal{I})$ and $t_j \in \mathcal{J}_c \in C(\mathcal{J})$ can always be found, so we have $\mathcal{P}_{\text{BSH}}^{\mathcal{I} \cup \mathcal{J}} = \mathcal{P}_{d_1}^{\mathcal{I} \cup \mathcal{J}} \geq \mathcal{P}_{\text{BSH}}^{\mathcal{I}_c \cup \mathcal{J}_c} = \infty$. $\qquad\square$

To prove that $\mathcal{P}_{\text{BSH}}^{\mathcal{I} \cup \mathcal{J}}$ pertains Non-prehensile & Non-vanishing, i.e., the non-prehensile and non-vanishing property, we also need to use induction. To this end, we establish the non-prehensile property for an index subset as follows:

**Definition A.2.** *The pairwise potential $\mathcal{P}_{BSH}^{\mathcal{I} \cup \mathcal{J}}$ pertains Non-prehensile & Non-vanishing restricted to $\langle \mathcal{I}, \mathcal{J} \rangle$ if, at every $x \in \mathcal{C}_{free}$, we can define a finite family of set pairs $\mathcal{A}_{\mathcal{I} \cup \mathcal{J}}(x)$ such that $\mathcal{P}_{BSH}^{\mathcal{I} \cup \mathcal{J}} = \sum_{\langle \mathcal{I}', \mathcal{J}' \rangle \in \mathcal{A}_{\mathcal{I} \cup \mathcal{J}}(x)} \mathcal{P}_{BSH}^{\mathcal{I}' \cup \mathcal{J}'}$, where every term $\mathcal{P}_{BSH}^{\mathcal{I}' \cup \mathcal{J}'}$ satisfy Equation 2. Further, for every pair of disjoint triangles $\langle t_i, t_j \rangle$ such that $t_i \in \mathcal{I}$ and $t_j \in \mathcal{J}$ or vice versa, we have $t_i \cup t_j \subseteq \mathcal{I}' \cup \mathcal{J}'$ for at least one $\langle \mathcal{I}', \mathcal{J}' \rangle \in \mathcal{A}_{\mathcal{I} \cup \mathcal{J}}(x)$.*

**Lemma A.3.** *If $\epsilon > 0$ then $\mathcal{P}_{BSH}^{\mathcal{I} \cup \mathcal{J}}$ pertains Non-prehensile & Non-vanishing restricted to $\langle \mathcal{I}, \mathcal{J} \rangle$ for any node pair $\mathcal{I}, \mathcal{J}$ of two rigid bodies.*

*Proof.* At $x \in \mathcal{C}_{\text{free}}$, we show the following two claims by induction from leaf to root:

- The pairwise potential $\mathcal{P}_{d_1}^{\mathcal{I} \cup \mathcal{J}} \geq \mathcal{P}_{\text{BSH}}^{\mathcal{I} \cup \mathcal{J}} \geq \mathcal{P}_{d_2}^{\mathcal{I} \cup \mathcal{J}}$ at any node when $\|x_{\mathcal{I}} - x_{\mathcal{J}}\| \geq R_{\mathcal{I}} + R_{\mathcal{J}}$.

- The pairwise potential $\mathcal{P}_{\text{BSH}}^{\mathcal{I} \cup \mathcal{J}}$ pertains Non-prehensile & Non-vanishing restricted to $\langle \mathcal{I}, \mathcal{J} \rangle$ for any node pair.

**Base Step:** For the pairwise potential $\mathcal{P}_{\text{BSH}}^{t_i \cup t_j} = \mathcal{P}_{d_1 \to d_2}^{t_i \cup t_j}$, we define $\mathcal{A}_{t_i \cup t_j}(x) = \{\langle t_i, t_j \rangle\}$, then Non-prehensile & Non-vanishing and the fact that $\mathcal{P}_{\text{BSH}}^{t_i \cup t_j} \geq \mathcal{P}_c^{t_i \cup t_j}$ follows from Corollary 5.2.

**Inductive Step I:** We assume our first claim holds for all $\mathcal{I}_c \in C(\mathcal{I})$ and $\mathcal{J}_c \in C(\mathcal{J})$, i.e. $\mathcal{P}_{\text{BSH}}^{\mathcal{I}_c \cup \mathcal{J}_c} \geq \mathcal{P}_c^{\mathcal{I}_c \cup \mathcal{J}_c}$ when $\|x_{\mathcal{I}_c} - x_{\mathcal{J}_c}\| \geq R_{\mathcal{I}_c} + R_{\mathcal{J}_c}$. We first show that $\mathcal{P}_{d_2}^{\mathcal{I} \cup \mathcal{J}} \leq \mathcal{P}_{d_1}^{\mathcal{I} \cup \mathcal{J}}$ when $\|x_{\mathcal{I}} - x_{\mathcal{J}}\| \geq R_{\mathcal{I}} + R_{\mathcal{J}}$. We note that $\mathcal{I} \cup \mathcal{J}$ contains at least 3 triangles, otherwise we reduce to the base case, so there are at least 2 terms of form $\mathcal{P}_{\text{BSH}}^{\mathcal{I}_c \cup \mathcal{J}_c}$. Each such term has the following lower bound:

$$\mathcal{P}_{\text{BSH}}^{\mathcal{I}_c \cup \mathcal{J}_c} \geq \mathcal{P}_c^{\mathcal{I}_c \cap \mathcal{J}_c} = 12 \left[ 1 + \frac{1}{\sqrt{\|x_{\mathcal{I}_c} - x_{\mathcal{J}_c}\|}} \right]^2 \geq 12 \left[ 1 + \frac{1}{\sqrt{\|x_{\mathcal{I}} - x_{\mathcal{J}}\| + R_{\mathcal{I}} + R_{\mathcal{J}}}} \right]^2 .$$

Here, the first inequality is due to our inductive condition and the fact that our BSH is a layered hierarchy by Definition A.2, so that $\|x_{\mathcal{I}} - x_{\mathcal{J}}\| \geq R_{\mathcal{I}} + R_{\mathcal{J}}$ implies $\|x_{\mathcal{I}_c} - x_{\mathcal{J}_c}\| \geq R_{\mathcal{I}_c} + R_{\mathcal{J}_c}$. The second inequality is because $x_{\mathcal{I}_c}$ (resp. $x_{\mathcal{J}_c}$) is at most $R_{\mathcal{I}}$ (resp. $R_{\mathcal{J}}$) from $x_{\mathcal{I}}$ (resp. $x_{\mathcal{J}}$). Using

the above lower bound, we derive the following estimate:

$$
\mathcal{P}_{d_1}^{\mathcal{I} \cup \mathcal{J}} \geq 24 \left[ 1 + \frac{1}{\sqrt{\|x_{\mathcal{I}} - x_{\mathcal{J}}\| + R_{\mathcal{I}} + R_{\mathcal{J}}}} \right]^2
$$
$$
\geq 24 + \frac{48}{\sqrt{\|x_{\mathcal{I}} - x_{\mathcal{J}}\| + R_{\mathcal{I}} + R_{\mathcal{J}}}} + \frac{24}{\|x_{\mathcal{I}} - x_{\mathcal{J}}\| + R_{\mathcal{I}} + R_{\mathcal{J}}}
$$
$$
\geq 24 + \frac{48}{\sqrt{2\|x_{\mathcal{I}} - x_{\mathcal{J}}\|}} + \frac{24}{2\|x_{\mathcal{I}} - x_{\mathcal{J}}\|}
$$
$$
\geq 12 + \frac{24}{\sqrt{\|x_{\mathcal{I}} - x_{\mathcal{J}}\|}} + \frac{12}{\|x_{\mathcal{I}} - x_{\mathcal{J}}\|} = 12 \left[ 1 + \frac{1}{\sqrt{\|x_{\mathcal{I}} - x_{\mathcal{J}}\|}} \right]^2 = \mathcal{P}_c^{\mathcal{I} \cup \mathcal{J}} = \mathcal{P}_{d_2}^{\mathcal{I} \cup \mathcal{J}},
$$

where we use the fact that $\|x_{\mathcal{I}} - x_{\mathcal{J}}\| \geq d_1 = R_{\mathcal{I}} + R_{\mathcal{J}}$ in the third inequality. As a result, we have our first claim holds for $\mathcal{I} \cup \mathcal{J}$ by the definition of the blending Equation 4.

**Inductive Step II:** We assume our second claim holds for all $\mathcal{I}_c \in C(\mathcal{I})$ and $\mathcal{J}_c \in C(\mathcal{J})$, i.e. $\mathcal{P}_{\mathrm{BSH}}^{\mathcal{I}_c \cup \mathcal{J}_c}$ pertains Non-prehensile & Non-vanishing restricted to $\langle \mathcal{I}_c, \mathcal{J}_c \rangle$. We show that $\mathcal{P}_{\mathrm{BSH}}^{\mathcal{I} \cup \mathcal{J}}$ pertains Non-prehensile & Non-vanishing restricted to $\langle \mathcal{I}, \mathcal{J} \rangle$ by considering three cases. Case II.a: If $\|x_{\mathcal{I}} - x_{\mathcal{J}}\| \leq d_1$, then $\mathcal{P}_{\mathrm{BSH}}^{\mathcal{I} \cup \mathcal{J}} = \mathcal{P}_{d_1}^{\mathcal{I} \cup \mathcal{J}}$ consists of terms of form $\mathcal{P}_{\mathrm{BSH}}^{\mathcal{I}_c \cup \mathcal{J}_c}$ each satisfying Non-prehensile & Non-vanishing by our inductive condition. Let us now define the finite family of set pairs by the union $\mathcal{A}_{\mathcal{I} \cup \mathcal{J}}(x) = \cup_{\mathcal{I}_c \in C(\mathcal{I}), \mathcal{J}_c \in C(\mathcal{J})} \mathcal{A}_{\mathcal{I}_c \cup \mathcal{J}_c}(x)$. It can be verified that this union is a disjoint union, and for every pair of disjoint triangles $\langle t_i \in \mathcal{I}, t_j \in \mathcal{J} \rangle$, we have $t_i \cup t_j \subseteq \mathcal{I}' \cup \mathcal{J}'$ belongs to exactly one such $\mathcal{A}_{\mathcal{I}_c \cup \mathcal{J}_c}(x)$. Further, we have $\mathcal{P}_{\mathrm{BSH}}^{\mathcal{I} \cup \mathcal{J}} = \sum_{\langle \mathcal{I}', \mathcal{J}' \rangle \in \mathcal{A}_{\mathcal{I} \cup \mathcal{J}}(x)} \mathcal{P}_{\mathrm{BSH}}^{\mathcal{I}' \cup \mathcal{J}'}$, where each $\mathcal{P}_{\mathrm{BSH}}^{\mathcal{I}' \cup \mathcal{J}'}$ satisfies Equation 2 due to Non-prehensile & Non-vanishing of the corresponding $\mathcal{P}_{\mathrm{BSH}}^{\mathcal{I}_c \cup \mathcal{J}_c}$ by our inductive condition. We have thus verified Non-prehensile & Non-vanishing of $\mathcal{P}_{\mathrm{BSH}}^{\mathcal{I} \cup \mathcal{J}}$. Case II.b: If $\|x_{\mathcal{I}} - x_{\mathcal{J}}\| \geq d_2$, then $\mathcal{P}_{\mathrm{BSH}}^{\mathcal{I} \cup \mathcal{J}} = \mathcal{P}_{d_2}^{\mathcal{I} \cup \mathcal{J}} = \mathcal{P}_c^{\mathcal{I} \cup \mathcal{J}}$ is a singled, centered potential. We trivially define $\mathcal{A}_{\mathcal{I} \cup \mathcal{J}}(x) = \{ \langle \mathcal{I}, \mathcal{J} \rangle \}$ then clearly every pair of disjoint triangles $t_i \cup t_j \subseteq \mathcal{I} \cup \mathcal{J} \in \mathcal{A}_{\mathcal{I} \cup \mathcal{J}}(x)$. Further, the induced force takes the following form:

$$
f_{i \in \mathcal{I}}^{\mathcal{I} \cup \mathcal{J}} = \frac{12(x_{\mathcal{I}} - x_{\mathcal{J}})}{|\mathcal{I}| \|x_{\mathcal{I}} - x_{\mathcal{J}}\|^{5/2}} \left[ 1 + \frac{1}{\|x_{\mathcal{I}} - x_{\mathcal{J}}\|^{1/2}} \right],
$$

which clearly belongs to $\mathcal{F}_{\mathcal{J} \to \mathcal{I}}$, thus we have verified all conditions in Non-prehensile & Non-vanishing of $\mathcal{P}_{\mathrm{BSH}}^{\mathcal{I} \cup \mathcal{J}}$. Case II.c: If $d_1 < \|x_{\mathcal{I}} - x_{\mathcal{J}}\| < d_2$, then $\mathcal{P}_{\mathrm{BSH}}^{\mathcal{I} \cup \mathcal{J}}$ is a blending of $\mathcal{P}_{d_1}^{\mathcal{I} \cup \mathcal{J}}$ in Case II.a and $\mathcal{P}_{d_2}^{\mathcal{I} \cup \mathcal{J}}$ in Case II.b, with strictly positive weights. We again trivially define $\mathcal{A}_{\mathcal{I} \cup \mathcal{J}}(x) = \{ \langle \mathcal{I}, \mathcal{J} \rangle \}$ to have every pair of disjoint triangles $t_i \cup t_j \subseteq \mathcal{I} \cup \mathcal{J} \in \mathcal{A}_{\mathcal{I} \cup \mathcal{J}}(x)$. The only condition we need to verify is that $\mathcal{P}_{\mathrm{BSH}}^{\mathcal{I} \cup \mathcal{J}}$ satisfies Equation 2 as a single term. We use a similar technique as in the proof of Lemma 5.1 by expanding the force term to get Equation 9 where there are three terms. For term II, We have by the analysis in case II.b that $-\partial \mathcal{P}_{d_2}^{\mathcal{I} \cup \mathcal{J}} / \partial x_{i \in \mathcal{I}} \in \mathcal{F}_{\mathcal{J} \to \mathcal{I}}$ and $\phi_{d_1 \to d_2}(x) > 0$ is strictly positive, so term II belongs to $\mathcal{F}_{\mathcal{J} \to \mathcal{I}}$. Term III is zero or belongs to $\mathcal{F}_{\mathcal{J} \to \mathcal{I}}$ because $\mathcal{P}_{d_1}^{\mathcal{I} \cup \mathcal{J}} \geq \mathcal{P}_{d_2}^{\mathcal{I} \cup \mathcal{J}}$ by our first claim. For term I, we know from the analysis in Case II.a that:

$$
-\frac{\partial \mathcal{P}_{d_1}^{\mathcal{I} \cup \mathcal{J}}}{\partial x_{i \in \mathcal{I}}} = \sum_{\langle \mathcal{I}', \mathcal{J}' \rangle \in \mathcal{A}_{\mathcal{I} \cup \mathcal{J}}(x)} -\frac{\partial \mathcal{P}_{\mathrm{BSH}}^{\mathcal{I}' \cup \mathcal{J}'}}{\partial x_{i \in \mathcal{I}}},
$$

where each term $-\partial \mathcal{P}_{\mathrm{BSH}}^{\mathcal{I}' \cup \mathcal{J}'} / \partial x_{i \in \mathcal{I}}$ is zero or belongs to $\mathcal{F}_{\mathcal{J}' \to \mathcal{I}'} \subseteq \mathcal{F}_{\mathcal{J} \to \mathcal{I}}$. Combined with the fact that the coefficient $(1 - \phi_{d_1 \to d_2}(x)) > 0$ is strictly positive, we conclude that term I is zero or belongs to $\mathcal{F}_{\mathcal{J} \to \mathcal{I}}$. As a result, we see that $\mathcal{P}_{\mathrm{BSH}}^{\mathcal{I} \cup \mathcal{J}}$ satisfies Equation 2 as a single term, so Non-prehensile & Non-vanishing holds. □

*Proof of Theorem 5.4.* Barrier-Form and Non-prehensile & Non-vanishing follows from Lemma A.1 and Lemma A.3, respectively. Smoothness follows from the fact that $\mathcal{P}_{\mathrm{BSH}}^{\mathcal{I} \cup \mathcal{J}}$ is derived by a finite number of blending between pairwise potentials, and all potentials and blending operators are twice differentiable. □

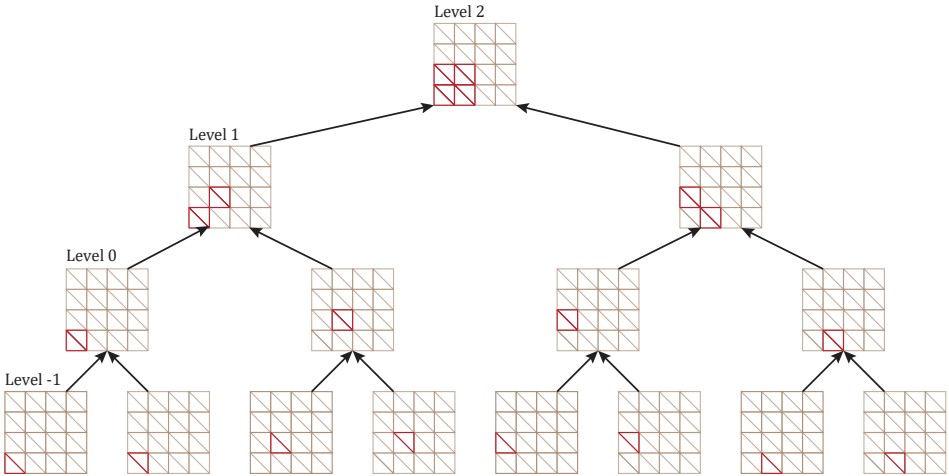

Figure 10: A special BSH constructed for the uniform grid. The levels of the BSH are indexed from bottom up. The $-1$th level contains only leaves. Every $(2i + 1)$th level ($i \geq 0$) merges two diagonal, rectangular blocks. Therefore, every $(2i)$th level consists of a cubic mesh block of side length $2^i$.

### A.2 COMPLEXITY ANALYSIS FOR A UNIFORM GRID

We show that, in the special case of two rigid bodies in the shape of a square uniform grid with infinitesimal distance to each other, as illustrated in Figure 11, the cost of evaluating $\mathcal{P}_{\text{BSH}}$ is $O(T)$. Here we assume a 2D uniform grid with $N^2$ grid cells so that $T = O(N^2)$. Without a loss of generality, we assume the grid size is 1. Further, for ease of analysis, we adopt a special construction of BSH as illustrated in Figure 10. It is easy to see that the bounding sphere radius of nodes in each level is all the same, which is denoted as $r_i$. We have the following results for estimating $r_i$, which can be derived directly from the construction of $\mathcal{P}_{\text{BSH}}^{\mathcal{I} \cup \mathcal{J}}$:

- $r_{-1} = \sqrt{5}/3 \quad r_0 = \frac{\sqrt{2} + 2\sqrt{5}}{6}$
- $r_{2i-1} = r_{2i} \leq 2^{i-1/2} C_r \quad \forall i > 0$ with $C_r = \frac{1 + \sqrt{10}}{3}$

To derive the second property, note that the tightest bounding sphere for the geometry of a $(2i)$-level node is $2^{i-1/2}$. This implies that the tightest bounding sphere for the geometry of a 0-level node is $1/\sqrt{2}$. However, the actual $r_0 = (\sqrt{1} + \sqrt{10})/3$, so the bounding sphere is unnecessarily scaled by $C_r$. By induction, we can prove that we can scale all tightest bounding spheres by $C_r$ accordingly to satisfy Definition 5.3.

To analyze the cost of evaluating $\mathcal{P}_{\text{BSH}}^{\mathcal{I} \cup \mathcal{J}}$, we note that, the cost reduces to evaluating a series of interaction terms either using centered potential ($\mathcal{P}_c^{\mathcal{I} \cup \mathcal{J}}$) or between leaf nodes ($\mathcal{P}^{t_i \cup t_j}$). Further by the recursive definition and the special structure of our BSH, the interaction terms are always computed between two nodes at the same level. Therefore, we can upper bound the number of interaction terms level by level.

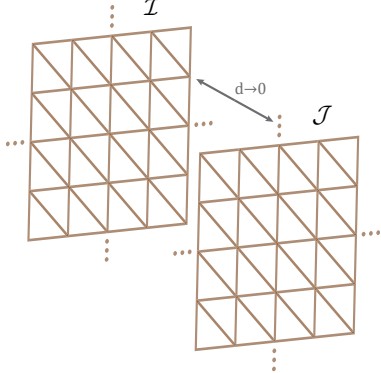

Figure 11: Two rigid bodies in the shape of a square uniform grid with infinitesimal distance $d \to 0$.

**Case I: $(2i)$-level Node** We focus on the nodes at $(2i)$th level, which bounds a cubic mesh block of side length $2^i$. We can denote these mesh blocks as $\mathbb{B}_{mn}^{2i}$ indexed using subscript $mn$. In other words, $\mathbb{B}_{mn}^{2i}$ consists of all triangles with coordinates within $[m2^i, (m + 1)2^i] \times [n2^i, (n + 1)2^i]$. For two

such blocks indexed by $\mathbb{B}^{2i}_{mn} \subset \mathcal{I}$ and $\mathbb{B}^{2i}_{m'n'} \subset \mathcal{I}$ on the two rigid bodies, we can define their distance as:

$$d(\mathbb{B}^{2i}_{mn}, \mathbb{B}^{2i}_{m'n'}) = \max(|m - m'|, |n - n'|).$$

Now suppose $d(\mathbb{B}^{2i}_{mn}, \mathbb{B}^{2i}_{m'n'}) \geq 2$, we can be sure that $\mathbb{B}^{2i}_{mn}$ and $\mathbb{B}^{2i}_{m'n'}$ belong to different $(2i+2)$-level blocks. Specifically, we define $\mathbb{B}^{2i}_{mn} \subseteq \mathbb{B}^{2i+2}_{\bar{m}\bar{n}}$ and $\mathbb{B}^{2i}_{m'n'} \subseteq \mathbb{B}^{2i+2}_{\bar{m}'\bar{n}'}$ and we have the following relationship:

$$d(\mathbb{B}^{2i+2}_{\bar{m}\bar{n}}, \mathbb{B}^{2i+2}_{\bar{m}'\bar{n}'}) \geq \left\lfloor \frac{d(\mathbb{B}^{2i}_{mn}, \mathbb{B}^{2i}_{m'n'})}{2} \right\rfloor. \tag{10}$$

Now let us define $\mathcal{I} = \mathbb{B}^{2i+2}_{\bar{m}\bar{n}}$ and $\mathcal{J} = \mathbb{B}^{2i+2}_{\bar{m}'\bar{n}'}$, we know that if the distance between the two cubic mesh blocks is sufficiently faraway, the potential term $\mathcal{P}^{\mathcal{I}\cup\mathcal{J}}$ reduces to the centered potential $\mathcal{P}^{\mathcal{I}\cup\mathcal{J}}_c$ can be computed via Equation 7 without utilizing any information from lower levels. A sufficient condition for this to happen is as follows:

$$d(\mathbb{B}^{2i+2}_{\bar{m}\bar{n}}, \mathbb{B}^{2i+2}_{\bar{m}'\bar{n}'}) 2^i \geq 2r_{2i+2}(1 + \epsilon). \tag{11}$$

Combining Equation 10 and Equation 11, we know that if $d(\mathbb{B}^{2i}_{mn}, \mathbb{B}^{2i}_{m'n'}) \geq 2\lceil 2\sqrt{2}C_r(1+\epsilon)\rceil$, then the interaction between the two $(2i)$-level blocks would be handled by the two $(2i+2)$-level super-blocks. Therefore, in order to compute the contact potential, we only need to evaluate the interaction between $B^{2i}_{mn}$ and at most $(1 + 4\lceil 2\sqrt{2}C_r(1+\epsilon)\rceil)^2$ blocks around it. The number of $(2i)$-level interaction terms of form $\mathcal{P}^{\mathcal{I}\cup\mathcal{J}}_c$ is:

$$O\left(\left\lceil \frac{N}{2^i} \right\rceil^2 (1 + 4\lceil 2\sqrt{2}C_r(1+\epsilon)\rceil)^2\right),$$

where the first part $\lceil N/2^i \rceil^2$ is the number of $(2i)$-level blocks and the second part is the number of other $(2i)$-level blocks, with which an interaction term $\mathcal{P}^{\mathcal{I}\cup\mathcal{J}}_c$ needs to be calculated.

**Case II: $(2i-1)$-level Node**  We have finished analyzing the cost of $(2i)$-level node interactions. The case with $(2i-1)$-level node iterations $(i \geq 1)$ is almost identical. Each $(2i-1)$-level node consists of half the triangles of a $(2i)$-level node, so we can assume the $(2i)$-level node $B^{2i}_{mn}$ has two children denoted as $\mathbb{B}^{2i-1,l}_{mn}$ and $\mathbb{B}^{2i-1,r}_{mn}$. Without a loss of generality, we only consider $\mathbb{B}^{2i-1,l}_{mn}$, which has the same bounding sphere center as that of $\mathbb{B}^{2i}_{mn}$. By the analysis of Case I, we know that, for all the children of $\mathbb{B}^{2i}_{m'n'}$ with $d(\mathbb{B}^{2i}_{mn}, \mathbb{B}^{2i}_{m'n'}) \geq 2\lceil 2\sqrt{2}C_r(1+\epsilon)\rceil$, their interaction with $\mathbb{B}^{2i-1,l}_{mn}$ would be taken care of at level $(2i+2)$. Again, we conclude that we only need to evaluate the interaction between $\mathbb{B}^{2i-1,l}_{mn}$ and the children of at most $(1 + 4\lceil 2\sqrt{2}C_r(1+\epsilon)\rceil)^2$ $(2i)$-level nodes around it. The number of $(2i-1)$-level interaction terms of form $\mathcal{P}^{\mathcal{I}\cup\mathcal{J}}_c$ is again:

$$O\left(\left\lceil \frac{N}{2^i} \right\rceil^2 (1 + 4\lceil 2\sqrt{2}C_r(1+\epsilon)\rceil)^2\right).$$

**Case III: $-1$-level Leaf Node**  The case with leaf nodes is exactly the same as that of $(2i-1)$-level nodes. Each 0-level node has two children at $-1$-level denoted as $\mathbb{B}^{-1,l}_{mn}$ and $\mathbb{B}^{-1,r}_{mn}$. Focusing on $\mathbb{B}^{-1,l}_{mn}$ and by the analysis of Case II, we know that, for all the children of $\mathbb{B}^0_{m'n'}$ with $d(\mathbb{B}^0_{mn}, \mathbb{B}^0_{m'n'}) \geq 2\lceil 2\sqrt{2}C_r(1+\epsilon)\rceil$, their interaction with $\mathbb{B}^{-1,l}_{mn}$ would be taken care of at level 2. Therefore, we conclude that we only need to evaluate the interaction between $\mathbb{B}^{-1,l}_{mn}$ and the children of at most $(1 + 4\lceil 2\sqrt{2}C_r(1+\epsilon)\rceil)^2$ 0-level nodes around it. The number of $(2i-1)$-level interaction terms of form $\mathcal{P}^{t_i\cup t_j}$ is again:

$$O\left(N^2(1 + 4\lceil 2\sqrt{2}C_r(1+\epsilon)\rceil)^2\right).$$

Put everything together, the cost of evaluating $\mathcal{P}_{\text{BSH}}$ is:

$$\sum_{i=0}^{\lceil \log_2 N \rceil} O\left(\left\lceil \frac{N}{2^i} \right\rceil^2\right) = O(N^2) = O(T).$$

### A.3 FRICTIONAL CONTACT MODELING

A feature-complete potential function should be able to handle frictional contacts. To this end, we adopt the technique proposed by Li et al. (2020); Ye et al. (2025) and consider the contact potential between the pair of triangles $\mathcal{P}^{t_i \cup t_j}(x_{i(1)}^t, x_{i(2)}^t, x_{i(3)}^t, x_{j(1)}^t, x_{j(2)}^t, x_{j(3)}^t)$, where we write explicitly the six vertices related to this potential function. The negative gradient norm of this potential is the normal force applied on the corresponding vertices. Li et al. (2020) proposes to model the contact potential as a tangential velocity damping term weight by the normal force magnitude. Put together, the frictional damping term takes the following form:

$$\mathcal{D}(x^{t+1}, x^t, \delta t) = \sum_{t_i \neq t_j} \left[ \sum_{k=1}^{3} \lambda \left\| \frac{\mathcal{P}^{t_i \cup t_j}}{x_{i(k)}^t} \right\| D_\parallel(x_{i(k)}^{t+1}, x_{i(k)}^t, \delta t) + \sum_{k=1}^{3} \lambda \left\| \frac{\mathcal{P}^{t_i \cup t_j}}{x_{j(k)}^t} \right\| D_\parallel(x_{j(k)}^{t+1}, x_{j(k)}^t, \delta t) \right],$$

where the summation is taken over triangle pairs on different rigid bodies, $D_\parallel$ is the tangential velocity damping term, which penalizes the relative velocity between $t_i$ and $t_j$, and $\lambda$ is the frictional coefficient. However, the above formulation is not strictly second-order differentiable. To fix this problem, Ye et al. (2025) proposed a novel definition of $D_\parallel$, which does not penalize the relative velocity between $t_i$ and $t_j$. Instead, they penalize the relative velocity between the two triangles and the separating plane $p_{ij}$, assuming the plane is a physical object with zero-mass. We refer readers to their work for more details.

The original frictional damping term can only deals with frictions between triangles. However, our contact potential $\mathcal{P}_{\text{BSH}}^{\mathcal{I} \cup \mathcal{J}}$ is a hierarchical blending of potentials between triangles and centered potentials. To extend the frictional damping term to use our $\mathcal{P}_{\text{BSH}}^{\mathcal{I} \cup \mathcal{J}}$, we propose to disregard the centered potentials and only consider potentials between triangles. Specifically, we propose the following potential between a pair of triangles:

$$\begin{cases} d_1 = R_{t_i} + R_{t_j} \text{ and } d_2 = (1 + \epsilon)d_1 \\ \mathcal{P}_{d_1}^{t_i \cup t_j} = \mathcal{P}^{t_i \cup t_j} \text{ and } \mathcal{P}_{d_2}^{t_i \cup t_j} = 0 \\ \mathcal{P}_{\text{local}}^{t_i \cup t_j} = \mathcal{P}_{d_1 \to d_2}^{t_i \cup t_j} \end{cases} . \tag{12}$$

Compared with the potential between leaf nodes in Equation 6, Equation 12 is not globally supported and vanishes when the distance between triangles is larger than $d_2$, denoted using subscript $\bullet_{\text{local}}$. This design choice would not cause gradient vanish because our normal contact potential $\mathcal{P}$ always provides non-vanishing gradient. In parallel, the benefit of using a locally supported function is that we can use a bounding volume hierarchy to quickly reject faraway triangles as done in Li et al. (2020); Ye et al. (2025). Equation 12 is plugged into the frictional damping term to yield our final formulation:

$$\mathcal{D}(x^{t+1}, x^t, \delta t) = \sum_{t_i \neq t_j} \left[ \sum_{k=1}^{3} \lambda \left\| \frac{\mathcal{P}_{\text{local}}^{t_i \cup t_j}}{x_{i(k)}^t} \right\| D_\parallel(x_{i(k)}^{t+1}, x_{i(k)}^t, \delta t) + \sum_{k=1}^{3} \lambda \left\| \frac{\mathcal{P}_{\text{local}}^{t_i \cup t_j}}{x_{j(k)}^t} \right\| D_\parallel(x_{j(k)}^{t+1}, x_{j(k)}^t, \delta t) \right].$$

Intuitively, our frictional damping model assumes that two triangles can only impose frictional damping forces on each other if their distance is less than $d_2$. Otherwise, the two triangles can only impose normal forces, but not frictional damping forces. This design choice preserves the property of non-vanishing gradient, but also allows efficient evaluation. Finally, we emphasize that as $\mu \to 0$, both our contact potential and frictional damping term converges to the exact frictional contact model with Coulomb friction.

## A.4 TWICE-DIFFERENTIABILITY OF IPC

We show that the IPC contact model (Li et al., 2020) is differentiable but not twice differentiable. Let us consider a simple 2D case, where the only geometric primitive pairs that incur collision potential is between a point and a line-segment. Let us now assume the toy example with a single geometric primitive: a line segment with two end points located at $(1,0)$ and $(2,0)$ and a point moving on the line $(x,1)$ with $x \in [0,3]$. The IPC potential for this toy example is formulated as: $\mathcal{P}(x) = -\log(d(x))$, where $d(x)$ is the distance between the point and the line-segment, parameterized by a single scalar $x$. Clearly, the differentiability of $\mathcal{P}$ relies on the differentiability of $d(x)$. As analyzed in Li et al. (2020), $d(x)$ is differentiable,

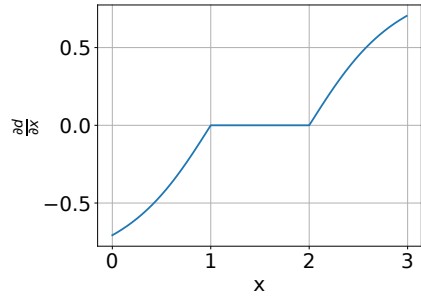

Figure 12: We plot the value of $\partial d(x)/\partial x$ when $x \in [0,3]$ in a toy example under the IPC contact model.

so $\partial d(x)/\partial x$ is well-defined. In Figure 12, we plot the value of $\partial d(x)/\partial x$ when $x \in [0,3]$. Clearly, $\partial d(x)/\partial x$ is not a differentiable function, so we conclude that $\mathcal{P}(x)$ cannot be twice-differentiable. The non-smoothness is due to switching between different Voronoi regions. When $x \leq 1$ or $x \geq 2$, the closest point on the line-segment is a vertex. Instead, when $x \in (1,2)$, the closest point lies interior to the line segment.

## A.5 HIERARCHICAL POTENTIAL BLENDING

As the same in Li et al. (2020); Ye et al. (2025), we use the Newton method equipped with line-search to solve the optimization problem of Equation 1, where we recursively accumulate the contact potential, its gradient, and Hessian based on BSH mentioned in Section 5 to accelerate the contact potential evaluation. Specifically, we establish a BSH for each joint in the scene and then recursively perform collision detection based on BSH while accumulating potential. When detecting a potential collision pair where nodes $\mathcal{I}$ and $\mathcal{J}$ are both leaf nodes, or the distance between two nodes is greater than $d_2$, it reaches the end of the recursion and return $\mathcal{P}^{\mathcal{I} \cup \mathcal{J}}_{d_1 \to d_2}$ and $\mathcal{P}^{\mathcal{I} \cup \mathcal{J}}_{d_2}$ respectively. Otherwise, we continue to search for their child nodes and blend it's potential using Equation 8. We summarize the algorithm of hierarchical potential blending in Algorithm 1.

---

**Algorithm 1** Hierarchical Potential Blending

---

**function** PROCESS_PAIR($\mathcal{I}, \mathcal{J}$)
    $\mathcal{P}^{\mathcal{I} \cup \mathcal{J}}_{d_1} \leftarrow 0$
    $\mathcal{P}^{\mathcal{I} \cup \mathcal{J}}_{d_2} \leftarrow \mathcal{P}^{\mathcal{I} \cup \mathcal{J}}_c$   ▷ *Equation 7*
    **if** $\|x_{\mathcal{I}} - x_{\mathcal{J}}\| > d_2$ **then**
        $\mathcal{P}^{\mathcal{I} \cup \mathcal{J}}_{\text{BSH}} \leftarrow \mathcal{P}^{\mathcal{I} \cup \mathcal{J}}_{d_2}$
        **return**
    **if** $\mathcal{I} = t_i$ is leaf and $\mathcal{J} = t_j$ is leaf **then**
        $\mathcal{P}^{\mathcal{I} \cup \mathcal{J}}_{d_1} \leftarrow \mathcal{P}^{t_i \cup t_j}$   ▷ *Section 4*
        Plug $\mathcal{P}_{d_1}, \mathcal{P}_{d_2}$ in Equation 7 for $\mathcal{P}_{d_1 \to d_2}$
        $\mathcal{P}^{\mathcal{I} \cup \mathcal{J}}_{\text{BSH}} \leftarrow \mathcal{P}^{\mathcal{I} \cup \mathcal{J}}_{d_1 \to d_2}$
        **return**
    **for** each pair $< \mathcal{I}_c, \mathcal{J}_c >$ in $< \mathcal{I}, \mathcal{J} >$'s children **do**
        PROCESS_PAIR($\mathcal{I}_c, \mathcal{J}_c$)
        $\mathcal{P}^{\mathcal{I} \cup \mathcal{J}}_{d_1} += \mathcal{P}^{\mathcal{I}_c \cup \mathcal{J}_c}_{\text{BSH}}$
    Plug $\mathcal{P}_{d_1}, \mathcal{P}_{d_2}$ in Equation 4 for $\mathcal{P}_{d_1 \to d_2}$
    $\mathcal{P}^{\mathcal{I} \cup \mathcal{J}}_{\text{BSH}} \leftarrow \mathcal{P}^{\mathcal{I} \cup \mathcal{J}}_{d_1 \to d_2}$
    **return**

---

## A.6 Experimental Details

In this section, we provide experimental details and extended evaluations.

| Parameter | Billiards | Push | Ant-Push | Sort | Gather | Gather-Bunny |
|---|---|---|---|---|---|---|
| trajectory horizon $H$ | 100 | 200 | 240 | 700 | 300 | 550 |
| receding horizon $h$ | / | 48 | / | 16 | 32 | 32 |
| potential coefficient $\mu$ | $1e^{-7}$ | $1e^{-6}$ | $1e^{-8}$ | $3e^{-8}$ | $3e^{-8}$ | $5e^{-9}$ |
| $\alpha$ learning rate | $3e^{-2}$ | $1e^{-2}$ | $3e^{-2}$ | $1e^{-2}$ | $1e^{-2}$ | $1e^{-2}$ |
| Adam $(\beta_1, \beta_2)$ | (0.3,0.5) | (0.3,0.5) | (0.3,0.5) | (0.3,0.5) | (0.3,0.5) | (0.3,0.5) |
| number of iterations | 400 | 50 | 100 | 100 | 60 | 60 |
| degrees of freedom | 96 | 12 | 22 | 66 | 66 | 66 |
| timesteps $\Delta t$ | 0.04 | 0.04 | 0.02 | 0.04 | 0.04 | 0.04 |

Table 3: Parameter settings for different benchmarks. Here, the trajectory horizon $H$ represents the total number of frames in the entire trajectory. For benchmarks requiring receding horizon execution, the receding horizon $h$ represents the number of frames in the sub-trajectory. The potential coefficient $\mu$ represents the contact energy coefficient, where a smaller $\mu$ indicates a more physically accurate contact mechanism. $\alpha$, the learning rate, denotes the step size for optimization, and the number of iterations specifies the total optimization steps. For all the benchmarks, we set the hyper-parameter of the Adam optimizer (Kingma & Ba, 2014), $\beta_1$ and $\beta_2$, to small values, which helps escaping from local minima. $\Delta t$ represents the timestep for each frame. Our simulator allows stable, penetration-free simulation even under relatively large timesteps.

### A.6.1 Baselines

We compare our method against three state-of-the-art baselines. The first is the IPC contact model (Li et al., 2020), employed in the differentiable simulator of (Huang et al., 2024). However, due to the model's lack of twice-differentiability, the implicit function theorem does not apply. As a result, Huang et al. (2024) resort to a few iterations of gradient descent to approximately solve Equation 1. While this yields usable gradient information, it is well known that the gradients can vanish when the interacting geometric primitives are not in close proximity. The second baseline is SDRS contact model (Ye et al., 2025), it's a differentiable model with twice-differentiability. But similar to IPC, the gradient also vanish when the interacting convex hulls are far apart. The third baseline is the Gradient Bundle (GB) method (Suh et al., 2022b), which addresses gradient vanishing through sampling, evaluated in practice via Monte Carlo methods. However, when primitives are far apart, the likelihood of sampling a non-zero gradient decreases significantly. Consequently, gradients obtained from GB can be both noisy and prone to vanishing with high probability.

### A.6.2 Benchmark Details

We implement a full-featured rigid body simulator based on our novel contact model. Each benchmark scenario includes controlled objects and target objects. The controlled objects are actuated using a built-in PD controller, and the objective across all benchmarks is to manipulate collisions and contacts to move the target objects to their designated spatial positions. In all experiments, we use the following loss to measure the progress of optimizers:

$$\text{ReLU}(\|x_{\text{COM}} - x^{\star}_{\text{COM}}\|^2 - \epsilon^2_{\text{target}}),$$

where $x_{\text{COM}}$ and $x^{\star}_{\text{COM}}$ are the position and desired position of the center of mass of some target object. $\epsilon_{\text{target}}$ is the error coefficient, indicating that tasks are considered successful for certain objects when they are within $\epsilon_{\text{target}}$ of the goal. The statistics of benchmarks are summarized in Table 3.

**Billiards** In this task, the indices of the two target balls and their target locations are randomly selected. The objective is to control a distinct red ball so that it strikes the target balls through contact, moving them to their respective target positions. Each ball has 6 degrees of freedom, resulting in a total system of 96 degrees of freedom. We only control the initial horizontal positions and velocities of the red ball, corresponding to 4 control dimensions.

**Push** In this benchmark, the task goal is to control a rod to push a box to the target region. The system consists of 2 objects with a total of 12 degrees of freedom. At each timestep, a continuous

6-dimensional control signal is generated to control the rod. The control signal at each timestep is obtained by solving a receding-horizon optimization problem.

**Ant-Push**   In this benchmark, our goal is to drive the ant robot to move and push the box to the target position. The ant consists of a base, four large legs, and four small legs. The base includes 3 translational degrees of freedom and 1 rotational degree of freedom, while the upper legs and lower legs are connected using ball joints and hinge joints, respectively. As a result, the ant has a total of 16 degrees of freedom in the kinematic state. Combined with the 6 degrees of freedom of the box, the system has a total of 22 degrees of freedom in the kinematic state, of which we can control 12 degrees of freedom in the ant's legs. We use 4 accumulated sine wave signals to parameterize our controller for each degree of freedom of the ant's legs as done in Hu et al. (2019). In this case, the decision variables are the amplitude, frequency, and phase of the sine waves.

**Sort**   In this benchmark, two types of cubes are mixed together on the ground. Use a rod to push each type of cube to its target location without mixing with each other. We set up 10 target cubes and one rod, so the system kinematic state has a total of 66 degrees of freedom and we can control the 3 translational degrees of freedom of the rod via PD controller.

**Gather**   In this task, 10 cubes are randomly put on the ground. Use a rod to collect all the objects into one area. Again the task has a total of 66 degrees of freedom and we control 3 translational and 1 horizontal rotational degrees of freedom of the rod. To further validate the efficiency of our method, we can handle objects with more complex geometries in this scenario. We replace the cubes with bunnies and successfully complete the gather task, we call this benchmark Gather-Bunny.

### A.6.3   Ablation Study of Contact Property

In Figure 3, we discuss the various properties that a contact model needs to possess. Here, we will analyze the impact of these different properties on the contact model's performance in simulation and policy learning. Collision models without Barrier-Form would allow intersection between objects. It is well known that the distance function between objects is non-smooth when the objects are in collision. Therefore, if we take away Barrier-Form, then Smoothness will be violated automatically. Non-prehensile requires that normal collision forces between objects are pushing them apart, instead of pulling them together. Almost all existing collision models satisfy this property. We conduct an ablation study on Non-vanishing. Since the only difference between our method and SDRS model (Ye et al., 2025) is the additional satisfaction of Non-prehensile, the results of this ablation study can be observed in the comparison between our method and SDRS across various benchmarks in Section 6. This demonstrates the necessity of satisfying this property for policy convergence when objects are far apart.

We note that it is possible to take away Smoothness alone and conduct an ablation study. To this end, we modify our Equation 3 to use Li et al. (2020) instead of Ye et al. (2025). In other words, we combine Li et al. (2020) with a globally supported barrier function instead of the original locally supported version. By doing so, we still have Non-vanishing but fails Smoothness. We tested the results on the Gather and Push benchmarks in Figure 13, showing that without Smoothness, the policy convergence speed is significantly slower or may even fail to converge completely.

### A.6.4   Influence of Stiffness and Timestep

It is well known that the stiffness of a system and the timestep $\delta t$ value can affect the evaluation of gradients. Here, we validate the sensitivity of our contact model to stiffness and timestep. We demonstrate the impact of stiffness and timestep on policy optimization by verifying the convergence of strategies on the Push task. In our contact model, stiffness is determined by the contact coefficient $\mu$, where a smaller $\mu$ corresponds to a stronger system stiffness. Firstly, we fix $\mu = 1e^{-7}$ and record the simulation time required for strategy convergence under different timesteps. We observe that strategies converge faster with larger timesteps. This is because, when using MPC for decision-making, a larger timestep results in fewer frames for the same horizon length, reducing the number of gradient multiplications and avoiding the issues of gradient explosion and vanishing gradients that are common in optimization problems. Since our method is unconditionally robust to timestep size, for a specific task, we should choose the largest timestep within a reasonable range to

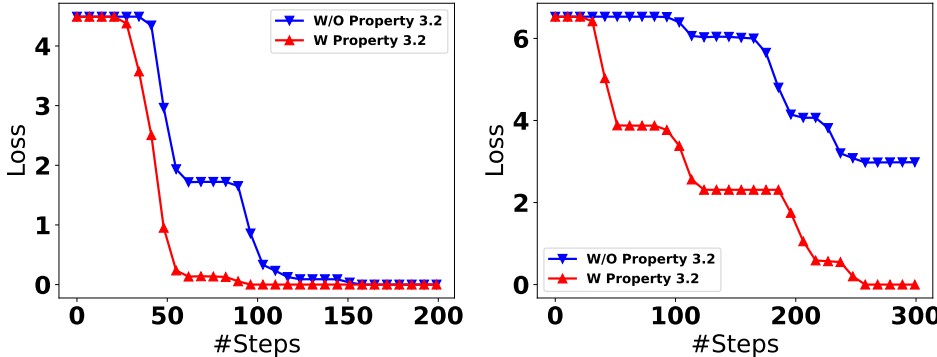

Figure 13: The convergency history with or without Smoothness on Push (left) and Gather (right) benchmark.

accelerate strategy convergence. Secondly, we fix $\delta t = 0.04s$ and record the simulation time required for strategy convergence under different $\mu$ values. We find that only when the system stiffness is extremely high ($\mu = 1e^{-10}$), leading to drastic gradient changes caused by stiffness, does the strategy convergence noticeably slow down. These results are shown in Table 4 and Table 5, respectively.

| Timestep $\delta t$ | $0.01s$ | $0.02s$ | $0.04s$ | $0.08s$ |
|---|---|---|---|---|
| Simulation Time to Converge | $5.76s$ | $3.28s$ | $2.96s$ | $2.80s$ |

Table 4: The simulation time required for policy convergence on the Push task under different $\delta t$.

| Contact Coefficient $\mu$ | $1e^{-6}$ | $1e^{-7}$ | $1e^{-8}$ | $1e^{-10}$ |
|---|---|---|---|---|
| Simulation Time to Converge | $3.20s$ | $3.28s$ | $3.24s$ | $4.80s$ |

Table 5: The simulation time required for policy convergence on the Push task under different $\mu$.

### A.6.5 COMPUTATIONAL EFFICIENCY

As described in Section 5, the brute-force evaluation of Equation 3 exhibits a time complexity of $O(T^2)$ with respect to the number of triangular facets, leading to rapidly increasing computational cost as the scene becomes more detailed. In contrast, our method achieves linear complexity in the special case outlined in Appendix A.2, with the only overhead stemming from nested optimization and hierarchical blending, compared to the IPC model (Li et al., 2020). To evaluate computational efficiency, we compared our method, brute-force computation, and IPC on the Billiards benchmark. We recursively subdivided the mesh of each ball to produce increasingly dense scenes, ranging from 512 to 6208 triangles in total. For each resolution, we performed full trajectory optimization to generate Figure 5. With brute-force computation, the average time per frame increased significantly from 3.35s to 604.50s. In contrast, our BSH-assisted contact model saw

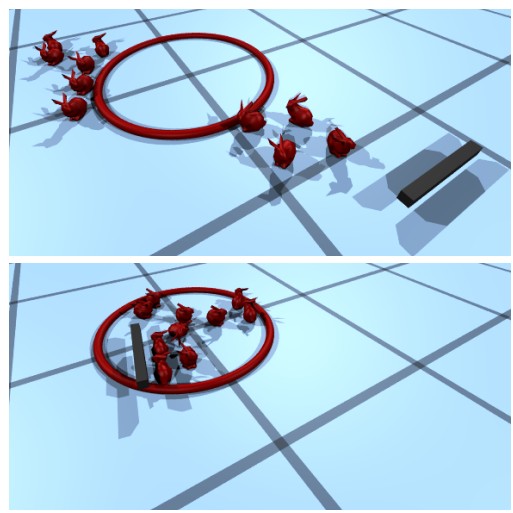

Figure 14: The start (top) and end (bottom) frame of the bunny-gather benchmark.

a modest increase from 0.6s to 3.18s. These results empirically confirm the expected quadratic growth of brute-force computation and the near-linear scaling of our method. Interestingly, the IPC model (Li et al., 2020) maintained a consistent per-frame cost of approximately 0.5s across all resolutions. This is attributed to the optimizer requiring fewer Newton iterations as mesh resolution increases, offsetting the higher evaluation cost of the contact model. We have also compared our performance using more complex, non-convex geometric shapes. As illustrated in Figure 14, we replace the cubes in our Gather benchmark with bunny meshes, with 2784 triangles in total. In this benchmark, named Gather-Bunny, we compare the average per-frame cost using brute-force computation, our method, and the IPC model, where the cost is 309.62s, 3.89s and 3.14s respectively. And in Figure 15, we have designed a more challenging articulated manipulation task, named Franka Push, where we use a 7-degree-of-freedom Franka robotic arm to push a bunny to a specified location. This task involves a total of 14343 triangles. We also compare the average per-frame cost using brute-force computation, our method, and the IPC model, with computating cost being >1000s, 7.83s, and 4.12s, respectively.

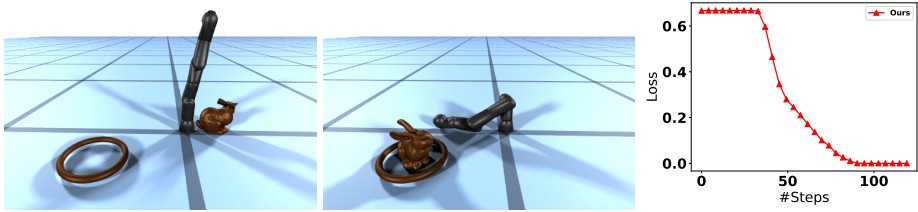

Figure 15: The start (left) and end (middle) frame of the Franka-Push benchmark. The convergency history is shown in the left.

