# OpenReview forum: "Efficient Differentiable Contact Model with Long-range Influence"
_ICLR.cc/2026/Conference — ICLR 2026 Poster_

### Official Review · Reviewer_5Qbn · 2025-10-29

**Soundness:** 3
**Presentation:** 2
**Contribution:** 2
**Rating:** 2
**Confidence:** 3

**Summary:**

In this paper, the authors formulate four properties a contact model must possess in order to have well-behaved gradients. Those properties are: 1. no interpenetration between contacting bodies, 2. smoothness, 3. unilateral normal forces, 4. non-vanishing gradients. The authors then propose a contact formulation based on potential functions for tedrahedal meshes that satisfies these properties. To increase the computational efficiency of their model, the authors propose to blend between different potentials, i.e., between the exact potential and an approximation based on bounding spheres. The authors evaluate their contact model for different control tasks in simulation and compare to two other contact models and randomized smoothing. The results show that their model provides more informative gradients and enables successful optimization.

**Strengths:**

-The paper rigurously motivates the proposed properties.

-The method (including the performance enhancements) is mathematically described in great detail.

-The results clearly show the gradient quality of the proposed contact model.

-The contact formulation using hierarchical potential blending is novel and computationally efficient.

**Weaknesses:**

-The claimed contribution of non-vanishing gradients has been studied several times before and is not novel. E.g. see [1,2]

-The baselines the contact model is compared to are limited and not representative of the relevant models in robotics. E.g. see [3,4]

-The supplementary videos should be presented with more effort, one could at least edit them together into a single video.

-My biggest concern with this paper is that physical accuracy is completely neglected. In the beginning, the authors acknowledge that there is a trade off between physical accuracy and well-behaved gradient information. However, they never investigate this trade off and solely focus on gradient quality in the experiments. In my opinion, very simple, smooth contact models can provide the same gradient quality achieved here with much less computation. The advantage of the proposed model could be increased physical realism but without experiments this is hard to judge. The impact and utility of the proposed method are very unclear.


[1] Turpin, Dylan, et al. "Grasp’d: Differentiable contact-rich grasp synthesis for multi-fingered hands." European Conference on Computer Vision. Cham: Springer Nature Switzerland, 2022.

[2] Schwarke, Clemens, et al. "Learning Deployable Locomotion Control via Differentiable Simulation." 9th Annual Conference on Robot Learning. 2025.

[3] Todorov, Emanuel, Tom Erez, and Yuval Tassa. "Mujoco: A physics engine for model-based control." 2012 IEEE/RSJ international conference on intelligent robots and systems. IEEE, 2012.

[4] Howell, Taylor A., et al. "Dojo: A differentiable simulator for robotics." arXiv preprint arXiv:2203.00806 9.2 (2022): 4.

**Questions:**

Thank you for the interesting paper! Maybe I misunderstand something here, but I was confused that the rod overcomes the box to push from the other side in the Push experiment. In my mind, even with contact gradients from a distance, the gradients from the side of the box that is closer to the rod in the beginning should always be stronger than those of the further side. What does the gradient look like if the contact surfaces are "behind" each other?

---

> ### Author Response · Authors · 2025-11-21
>
> **Q: The claimed contribution of non-vanishing gradients has been studied several times before and is not novel. E.g. see [1,2]**
>
> **Q: Very simple, smooth contact models can provide the same gradient quality. The advantage of the proposed model could be increased physical realism but without experiments this is hard to judge.**
>
> **A:** Our contribution is not merely addressing the non-vanishing gradient issue, but rather proposing 4 essential properties of a theoretically robust and universal contact model. We further design a contact model that satisfies all 4 properties, and demonstrate its strong performance across various locomotion and manipulation tasks. Instead, methods [1,2] use soft contact penalties, which satisfy properties 3.3a and 3.3b, but fail properties 3.1 and 3.2 since they allow penetration to happen. Although they are practical in certain tasks, they are not theoretically robust and universal contact models. Please refer to our revised Table 1.
>
> |     Formulation    | 3.1 | 3.2 |   3.3a   | 3.3b |
> |:-----------------:|:---------:|:------------------:|:--------:|:--------:|
> |        [1,2]       |  ❌  |        ❌       |   ✔️   |  ✔️   |
>
> We believe that [3,4] and [Li et al. 2020] belong to the same type of contact model that fails property 3.3a. Meanwhile, there are many other similar contact models widely used in robotics, such as [5,6]. While we believe it is redundant to compare with every contact model, we have added comparisons with MuJoCo in the revised figs 5,7,8,9 in Section 6.
>
> **Q: The supplementary videos should be presented with more effort, one could at least edit them together into a single video.**
>
> **A:** We consolidated all supplementary materials into a single video. Please refer to our latest supplementary materials for more details.
>
> **Q: physical accuracy is completely neglected**
>
> **A:** We agree that adjusting the contact coefficient $\mu$ allows for a trade-off between physical accuracy and gradient quality. In fact, in all experiments conducted in this paper, the maximum $\mu$ we selected was no greater than $1e^{-6}$, at which point the contact model provides effective gradients while introducing negligible physical errors. As cited in reviewer h8Nt's response: "That said, in the videos supplied with the supplementary material, I could not observe such artifacts."
>
> Even so, we have designed an experiment to further verify physical accuracy. We validate this using the book stacking problem [7] with a theoretical analytical solution. Our contact model can reproduce this experiment perfectly when $\mu$ is less than $1e^{-6}$, with error summarized in the table below. We recommend the reviewer to read section 6 and the fig 4 of the revised paper for more information.
>
> |     Contact Coefficient $\mu$    | 1e-5 |   1e-6   | 1e-7 | 1e-8 | 1e-10 |
> |:-----------------:|:---------:|:------------------:|:--------:|:--------:|:--------:|
> |       margin/m    |      1.82e-2         |   5.67e-3   | 1.47e-3    |   4.12e-4      |  3.13e-5    |
> |       force/N           |       6.38e-3        |  6.40e-4    |  6.42e-5   |   6.42e-6      |  6.42e-8    |
> |       suc           |       ❌        |  ✔️    |  ✔️   |    ✔️     |  ✔️    |
>
> Table: The margin between the planks, the error in the contact force received by the top plank, and the stability of the system, all with respect to the contact coefficient $\mu$.
>
>
> **Q: Why does the rod overcome the box to push from the other side?**
>
> **A:** Even if the rod is positioned in front, it will receive a gradient to move above, i.e. out of the way of the box. This is because moving the rod above will decrease the component of the virtual contact force acting on the front of the box, getting the box closer to the target position. As a result, the optimizer would automatically converge to a strategy of first moving the rod to the side of the box, then moving around to the back to push the box.
>
> **Q: What does the gradient look like if the contact surfaces are "behind" each other?**
>
> **A:** The contact faces will be numerically optimized to be facing each other, instead of behind each other. Due to strict convexity of the contact potential energy in terms of the faces, the aforementioned property is theoretically proven in [Ye et al., 2024]. Since we build our method on top of [Ye et al., 2024], this nice property is inherited in our method. In summary, the situation of "the contact surfaces are behind each other." is provably impossible.
>
> **[5]** Tao Pang, et al. "Global planning for contact-rich manipulation via local smoothing of quasi-dynamic contact models." IEEE Transactions on robotics, 2023.
>
> **[6]** J. Lee, et al. “DART: Dynamic animation and robotics toolkit,” Journal of Open Source Software, vol. 3, no. 22, p. 500, 2018.
>
> **[7]**  J. F. Hall, “Fun with stacking blocks,” American Journal of Physics, vol. 73, no. 12, pp. 1107–1116, 2005.

---

### Official Review · Reviewer_WJTs · 2025-10-31

**Soundness:** 3
**Presentation:** 2
**Contribution:** 2
**Rating:** 4
**Confidence:** 3

**Summary:**

This paper argues that the choice of contact models for resolving collisions plays a significant role in the quality of gradients computed by differentiable rigid body simulators. They establish necessary properties that a well-behaved contact model should possess, i.e. log-barrier, smoothness, non-prehensile and non-vanishing. The authors then propose a practical contact model which follows all of these properties, and further modify it to improve upon efficiency, while ensuring that the modified contact model is still well-behaved. The proposed method is demonstrated on a variety of gradient based optimization tasks, each of which involves learning a sequence of control signals on controlled objects, which when interacting with other target objects through the proposed contact model, result in a desired motion through simulation. Time complexity analysis is also provided for select tasks.

**Strengths:**

•	The theoretical contributions of the paper are quite dense and robust, and they cover all aspects of the proposed methodology in detail, with relevant proofs in the appendix.
	•	The tasks in the provided demos are quite varied and the corresponding results appear to be convincing.

**Weaknesses:**

- The exposition can be heavy in notation sometimes, making it hard to understand.
- Although the authors provide theoretical comparisons with other contact models, it would be even better if the comparison can be extended to the demos in the evaluation section. This might provide valuable insights on the importance of each of the properties described in Section 3.
- The authors might want to provide an overview of the methodology in terms of an algorithm pseudocode and/or implementation details, for exposition and reproducibility.

NOTE:   I reviewed an earlier version of this paper.  Most of the text including typos in figures and equations are exactly the same as last time.   I'd like to ask the authors to please at least correct these typos from previous reviews.

**Questions:**

•	Typo: In Figure 1, the caption should describe a hexagon, while it currently describes a pentagon
	•	In Section 4, in the expression for computing the contact potential between two triangles, should the RHS be (min L) instead of (argmin L)?
	•	What do you use for the value of mu, the coefficient of the contact model P?
	•	When the set of vertices are far enough, i..e. at distance d_2 in the notation of the paper, can we simply use the closed form centered potential as in eqn. (5), instead of blending it with the exact potential at d_1? This might be feasible considering the centered potential satisfies all the properties of a well-behaved contact potential as argued in the proof of Corollary 5.2. The general question here is: what is the motivation behind using blending?

---

> ### Author Response · Authors · 2025-11-21
>
> **Q: The authors might want to provide an overview of the methodology in terms of an algorithm pseudocode and/or implementation details.**
>
> **A:** In the revised paper's appendix A.5, we have revised the pseudocode and provided an overall algorithm description. Please refer to it for details.
>
> **Q: When the set of vertices are far enough, can we simply use the closed form centered potential instead of blending it?  What is the motivation behind using blending?**
>
> **A:** For the first question, that's correct. When the two vertex sets are far apart, energy mode of equation 5 will be used, which penalizes exactly the center distance. However, when the distance between the vertex sets is less than $d_1​$, we need to respect the exact geometry of objects by using the more exact potential energy $\mathcal{P}^{t_i\cup t_j}$.
>
> Now the reason for blending is for switching between the two energy modes above in a smooth manner to satisfy property 3.2. Further, we need to carefully design the blending function to also satisfy property 3.3a, which avoids attractive forces and ensures non-prehensile interactions.
>
> Another reason for blending is to utilize the BSH to accelerate the computation. As shown in our appendix A.2, in an idealized setting, the cost of evaluating our contact potential is linear in the number of triangles in the object. Such performance is not possible without blending.
>
> **Typo**
>
> **A:** Typos in fig 1 and section 4,5 are fixed in revised paper. For the value of $\mu$, please refer to the original paper’s table 2 in appendix A.6.

---

### Official Review · Reviewer_h8Nt · 2025-10-31

**Soundness:** 4
**Presentation:** 2
**Contribution:** 3
**Rating:** 8
**Confidence:** 4

**Summary:**

Section 2 provides a brief overview on differential simulators in the field of computer graphics. In Section 3, the authors discuss several desirable properties for gradient-based optimization of differentiable simulators. In particular, the authors point out that a contact potential should **approach infinity when the signed distance approaches zero** (aka follow a "log-barrier"), be **twice-differentiable** (aka "smooth"), **only create pushing forces** (aka "non-prehensible"), and **provide non-zero gradients** even if geometries do not touch ("non-vanishing"). In **Section 4**, the authors extend the contact potential to fulfill several desirable properties for gradient-based optimization of differentiable simulators. In particular, the authors modify the contact potential function of (Liang et al., 2024; Ye et al., 2024) to have global support. However, computing the proposed potential function
P is not efficient as it requires computing the contact potential between each pair of
disjoint triangles **increasing computation quadratically with the number of triangles**. Therefore, the authors suggest in **Section 5**  to interpolate between two potential function as a function of signed distance. If the objects are close the previously derived potential function is used while if the objects are far apart then a faster to compute potential is evaluated that uses a bounding sphere hierarchy to make  the algorithm computational tractable. In **Section 6** the authors show numerous optimizations of different simulation scenes involving simulator gradients.

**Strengths:**

The method developed for smoothly interpolating between different force potentials is quite clever and will definitely inspire follow up work in the field. The derivation of the potential functions appears to be sound.

- Figure 2 is quite nice.
- Nice experiments (though "only" in simulation).

**Weaknesses:**

- **C1: The notation is a bit convoluted and the writing could be clearer.**
  Some suggestions:
	- You could simply define index $i$ to belong to vertices I and $j$ to vertices J, then you have $f_i$ instead of $f_{i \in I}^{I \cup J}$ and $x_i \in x_{i \in I}$. Also you could define $\mathcal{P}^{i \cup J}$ instead of $\mathcal{P}$ (as it is clear that the potential function acts between sets of vertices).
	- I am not sure what Definition 3.1 is defining and why it required the use of a "Definition" statement.
	- "Lemma 4.1" seems also unnecessary. Also why define properties through numbers (e.g. Property 3.1, 3.2, 3.3, 3.4)?
     - "P(x)=\infty iff x \in C_obs" seems imprecise/wrong.

- **C2: Adding naive "global support" of the contact potential degrades simulation realism.** If you extend the force potential to provide global support then you apply contact forces in the simulation between object that do not touch. While these forces are small, they may notably alter the simulation. For simulation of real-world scene, extending global support could introduce a reality gap and adversely effect control synthesis. That said, in the videos supplied with the supplementary material,  I could not observe such artifacts. That said, the arxiv paper [Hard Contacts with Soft Gradients: Refining Differentiable Simulators for Learning and Control](https://arxiv.org/abs/2506.14186) might be interesting to the authors (this is an arxiv paper so there is no need to discuss it in the author's paper). In principle, the authors could use straight-through estimation to only use the contact potential with global support for the gradient computation.

**Questions:**

- Q1: The paper https://github.com/taichi-dev/difftaichi (not cited by the authors) discusses how time-discretization causes gradient errors for stiff contact simulations. How does increasing contact stiffness or integration times affect the gradient quality of the proposed simulator?

---

> ### Author Response · Authors · 2025-11-21
>
> **Q: You could define index $i$ to belong to vertices $\mathcal{I}$ and $j$ to vertices $\mathcal{J}$, then you have $f_i$ instead of $f_{i\in \mathcal{I}}^{\mathcal{I}\cup\mathcal{J}}$ and $x_i\in x_{i\in\mathcal{I}}$. Also you could define $\mathcal{P}^{i\cup\mathcal{J}}$ instead of $\mathcal{P}$.**
>
> **A:** We politely argue that our definition is more appropriate. First, $f_i$ with a single subscript is not enough because we need to specify two sets, $\mathcal{I}$ and $\mathcal{J}$, to indicate that the force is pointing from vertices in $\mathcal{I}$ to vertices in $\mathcal{J}$. Over our construction of BSH, both sets $\mathcal{I}$ and $\mathcal{J}$ could change. The same rule applies to $\mathcal{P}^{\mathcal{I}\cup\mathcal{J}}$. In our paper, we use both $\mathcal{P}^{\mathcal{I}\cup\mathcal{J}}$ and $\mathcal{P}$ without superscripts, where the later indicates the summation of potential terms over all distinct $\mathcal{I}$ and $\mathcal{J}$ pairs, which is defined in property 3.3.
>
> **Q: What is Definition 3.1 defining?**
>
> **A:** Note that property 3.1 and 3.2 are defined for the entire contact potential $\mathcal{P}$. For completeness, we need to restate these properties for pairwise contact potential $\mathcal{P}^{\mathcal{I}\cup\mathcal{J}}$. Definition 3.1 serves precisely this purpose.
>
> **Q: Lemma 4.1 seems unnecessary.**
>
> **A:** The proof of Lemma 4.1 is non-trivial and we refer the reviewer to appendix A.1.
>
> **Q: Why define properties through numbers?**
>
> **A:** We are using Latex's default theorem counter. We agree with the reviewer that using property names is clearer. We have revised our paper to use this format.
>
> **Q: "$P(x)=\infty$ iff $x \in C_{obs}$" seems imprecise/wrong.**
>
> **A:** This statement conveys that the contact potential $P(x)$ becomes infinite whenever the configuration $x$ contains a collision or penetration, and remains finite otherwise. We believe this formulation is precise. If the reviewer feels it is incorrect or misleading, we would be grateful for a suggested revision.
>
> **Q: Adding naive "global support" of the contact potential degrades simulation realism**
>
> **A:** We agree that using global support only for gradient computation is a good solution. However, as mentioned in review, this phenomenon was not observed in the demo. This is because in all benchmarks, we used a very small contact coefficient ($\mu < 1e^{-6}$), which was sufficient to provide ideal gradient information while the resulting virtual forces were almost negligible. To further elaborate, we supplemented this with a physical experiment on the book stacking problem [1] to verify the accuracy of the contact forces. We recommend the reviewer to read our revised section 6 and fig 4.
>
> **Q: How does contact stiffness & integration time affect the gradient quality?**
>
> **A:** We have supplemented the performance results regarding stiffness and integration times in the Push task in our revised appendix A.6.4. The optimizer tends to converge more easily for this task when the timestep $\delta t$ is larger. This is because, when using MPC for decision-making, a larger timestep results in fewer frames for the same horizon length, reducing the number of gradient multiplications and avoiding the issues of gradient explosion and vanishing that are common in optimization problems. Since our method is unconditionally stable under timestep size, for a specific task, we should choose the largest timestep within a reasonable range to accelerate strategy convergence. We also compared the convergence under different stiffness levels and found that, only when the contact stiffness is extremely low ($\mu=1e^{-10}$), the convergence speed suffers from a sharp drop. This implies the importance of sufficiently large gradient information when objects are not exactly in contact, reiterating the importance of property 3.3b.
>
> |     $\delta t$   | 0.01 |   0.02   | 0.04 |0.08|
> |:-----------------:|:---------:|:------------------:|:--------:|:--------:|
> |       Simulation time to converge (s)            |      5.76         |   3.28   | 2.96    |    2.80     |
>
> Table: The simulation time required for policy convergence on the Push task under different $\delta t$.
>
> |    $ \mu$    | 1e-6 |   1e-7   | 1e-8 |1e-10|
> |:-----------------:|:---------:|:------------------:|:--------:|:--------:|
> |       Simulation time to converge (s)            |       3.20        |   3.28   |  3.24   |  4.80       |
>
> Table: The simulation time required for policy convergence on the Push task under different $\mu$.
>
> **[1]**  J. F. Hall, “Fun with stacking blocks,” American Journal of Physics, vol. 73, no. 12, pp. 1107–1116, 2005.

---

### Official Review · Reviewer_sWeD · 2025-11-01

**Soundness:** 4
**Presentation:** 4
**Contribution:** 4
**Rating:** 8
**Confidence:** 3

**Summary:**

This paper presents a novel differentiable contact model for rigid-body simulation that guarantees both physical realism and well-behaved gradients. The authors identify key properties - log-barrier, smoothness, non-prehensility, and non-vanishing gradients as necessary for reliable gradient-based optimization in differentiable physics. They propose a new contact potential formulation that satisfies all these properties and introduce a Bounding Sphere Hierarchy (BSH) mechanism for computational efficiency, reducing the cost of evaluating contact interactions. Extensive experiments across contact-rich tasks demonstrate that this model produces smoother gradients, faster convergence, and improved stability compared to existing baselines.

**Strengths:**

1. **Strong theoretical foundation** - the work is grounded in rigorous mathematical analysis, formally defining the necessary conditions for a “well-behaved” contact potential. The authors’ proofs (Appendix A.1–A.3) are precise and significantly strengthen the paper’s technical depth.
2. **Novel and timely contribution** - differentiable contact modeling remains a bottleneck in differentiable simulation. Introducing a model that satisfies both smoothness and non-vanishing gradient properties is highly relevant, especially as differentiable simulators gain traction in RL and robotics (see Werling et al., 2021; Xu et al., 2022; Ye et al., 2024).
3. **Thorough experimental validation** - the experiments are well-designed and comprehensive, spanning both manipulation and locomotion tasks. The inclusion of both trivial and random initialization demonstrates robustness.
4. **High-quality writing and presentation**  -the manuscript is clear, well-organized, and logically develops from theoretical properties to practical implementation.
5. **Impact potential** - the long-range gradient property directly addresses a key failure mode in current differentiable physics pipelines (gradient vanishing in contact-free configurations), with broad implications for differentiable MPC, co-design, and world-model-based RL.

**Weaknesses:**

1. **Terminology ambiguity** - the use of “log-barrier” as a property (Property 3.1) is somewhat confusing. In optimization literature, log-barrier refers to a method, not an inherent property. Reframing this as a “barrier potential” or “barrier-form property” would avoid conceptual conflation.
2. **Referencing and clarity**
    * Propositions 3.3a and 3.3b are not explicitly labeled in Table 1, which may confuse readers unfamiliar with the notation.
    * The color scheme in Figure 2 is not self-explanatory; a legend or caption clarification would improve readability.
    * Figure 3 is visually appealing but does little to elucidate the core algorithmic contribution; it could be condensed or better explained.
3. **Accessibility for newcomers** - since the proposed formulation heavily builds on Liang et al. (2024) and Ye et al. (2024), a brief, self-contained overview of those works would make this paper more approachable.
4. **Presentation details** - several figures (especially in Sections 5–6) are too small to read comfortably. As a general guideline, the figure text size should match the main body font.
5. **Limited domain variety** - while the manipulation and locomotion experiments are strong, an articulated manipulation case (e.g., as in Srinivasan et al., 2024) would better demonstrate real-world robotic applicability.

**Questions:**

1. What does "GD" baseline refer to in Figures 4,6,7, and 8?
2. Given Figure 5, could the authors quantify how their method scales to mesh complexities in realistic robots (>10k triangles)? Is RL or trajectory optimization still feasible?
3. The method currently applies to rigid-body systems. Do the authors foresee any principled way to extend the non-vanishing gradient property to deformable or soft-body interactions?
4. The hierarchical blending (Eq. 8) relies on ϵ > 0 to guarantee theorems. How sensitive are results to this choice in practice?

---

> ### Author Response · Authors · 2025-11-21
>
> **Q: The use of “log-barrier” as a property (Property 3.1) is somewhat confusing.**
>
> **A:** We agree and have changed the name of property 3.1 to Barrier-Form in the revised paper.
>
> **Q: Propositions 3.3a and 3.3b are not explicitly labeled in Table 1.**
>
> **A:** We have referred to every property by name in table 1 of our revised paper.
>
> **Q: The color scheme in Figure 2 is not self-explanatory.**
>
> **A:** We have modified our caption to color corresponding texts using the same color as in the figure, serving as a legend.
>
> **Q: Figure 3 could be condensed or better explained.**
>
> **A:** Figure 3 is already presented as a condensed inset and is accompanied by a detailed caption. If the reviewer believes that additional clarification would be helpful, we would appreciate specific suggestions on how to improve the explanation.
>
> **Q: Authors should provide a brief, self-contained overview of [Ye et al. 2024].**
>
> **A:** Thanks to the one page extension of camera ready paper, we can provide such an overview in section 4 of our revised paper.
>
> **Q: The figure text size should match the main body font.**
>
> **A:** This was due to space constraints. Since the camera-ready version allows for an additional page, we have re-adjusted the layout. You can see the corrections to the figures in the revised paper.
>
> **Q: Could the authors quantify how their method scales to mesh complexities in realistic robots (>10k triangles)?**
>
> **A:** We have added a new example of a Franka robot pushing a bunny, which contains a total of 14343 triangles. We calculate the computational cost in appendix A.6.5 and the convergence curve can be found in revised fig 15. We demonstrate that the algorithm still works for articulated manipulation involving a large number of triangles.
>
> **Q: What does the “GD” baseline refer to in Figures 4,6,7, and 8?**
>
> **A:** This is actually “GB” instead of “GD”.“GB” stands for gradient bundle method. In this baseline, we use run accurate forward simulation but replace exact gradient with an randomized sampled smooth gradient during backward optimization. We refer the reviewer to [Suh et al. (2022b)] for more details.
>
> **Q: Can the method be extended to soft-body simulation?**
>
> **A:** We have discussed the possibility of such extension in our conclusive section 7. One issue with such extension is that our potential energy requires a nested optimization, which must be performed between each pair of nearby geometric primitives, which could be computationally demanding. Another issue is that our BSH is precomputed for rigid bodies, which must be recomputed when deformation happens. However, the non-prehensile property might be violated when such recomputation happens. Therefore, a direct extension could be difficult and we leave such extension to future works.
>
> **Q: How sensitive are results to the choice of $\epsilon$**
>
> **A:** An extremely small $\epsilon$ can lead to non-smooth gradients, while an excessively large $\epsilon$ can introduce excessive computational costs. Therefore, an appropriate $\epsilon$ needs to be chosen. However, according to the study in this paper, the algorithm is not sensitive to this parameter. For all benchmarks in this paper, we uniformly set $\epsilon=0.1$, which performs well across all benchmarks.

---

### Author Response · Authors · 2025-11-21

We sincerely thank all the reviewers for your recognition of the innovation in our paper, and the valuable feedback. We have submitted the revised paper, incorporating writing modifications based on your suggestions and adding new experimental results. Additionally, we have updated the supplementary material, which includes new results. We have contrasted our method with other baselines and consolidated all demos into one single video. We fix the citation of SDRS contact model from [Ye et al. 2024] to [Ye et al. 2025] in the revised paper. Here is a list of our new results:

**1** Addition of experiments validating the physical accuracy, detailed in section 6 and fig 4 of the revised paper.

**2** Comparison results with MuJoCo for the benchmarks discussed in the main text, illustrated in figs 5, 7, 8, and 9 of the revised paper.

**3** Ablation study on policy convergence with respect to contact stiffness and timestep size, elaborated in appendix A.6.4.

**4** Addition of an articulated manipulation benchmark with complex meshes, detailed in appendix A.6.5.

---

### Meta-Review · Area_Chair_tg7m · 2026-01-07

**Summary:**

This paper proposes a differentiable rigid-body contact model to produce well-behaved gradients for gradient-based optimization. The key idea is to identify desirable properties such as smoothness and non-vanishing gradients, and introduce an efficient implementation via hierarchical blending with a bounding-sphere hierarchy. It reached a consensus that the paper contains strong theoretical framing and detailed proofs, complemented by a practical design for efficiency. The manuscript is also overall well-written. There were some concerns regarding the lack of implementation clarity, the sufficiency of empirical comparisons, as well as the gap to real-world settings. The rebuttal has substantially addressed the major concerns, especially those regarding missing experiments and missing clarity. I suggest the authors incorporate all the feedback from the reviewers in preparing the camera-ready version of the paper.

**Reviewer Concerns:**

Concerns regarding naming/terminology, exposition and presentation, the gap to real-world settings, the clarity on certain algorithm details, and the comparison with sufficient baselines, have been properly addressed by the rebuttal. One key remaining concern is the relatively limited novelty compared to existing work on "non-vanishing gradients".

**Reviewer Scores:**

Reviewers sWeD and h8Nt who already gave 8 will likely to keep the very positive rating. Reviewers WJTs and 5Qbn will likely increase the score, as the rebuttal explicitly addressed the issues on clarity, presentation, and the comparison to baseline.

---

### Decision · Program_Chairs · 2026-01-26

Accept (Poster)